# The potential of increasing man-made air pollution to reduce rainfall over southern West Africa

Gregor Pante[1,2], Peter Knippertz[1], Andreas H. Fink[1], and Anke Kniffka[1,3]

[1]Institute of Meteorology and Climate Research, Department Troposphere Research (IMK-TRO), Karlsruhe Institute of Technology (KIT), Wolfgang-Gaede-Str. 1, 76131 Karlsruhe, Germany
[2]now at German Meteorological Service, Frankfurter Str. 135, 63067 Offenbach am Main, Germany
[3]now at German Meteorological Service - Research Centre Human Biometeorology, Stefan-Meier-Str. 4, 79104 Freiburg, Germany

**Correspondence:** Gregor Pante (gregor.pante@dwd.de)

**Abstract.** Southern West Africa has one of the fastest growing populations worldwide. This has led to a higher water demand and lower air quality. Over the last three decades, most of the region has experienced decreasing rainfall during the little dry season (LDS, mid-July to end of August) and more recently also during the second rainy season (SRS, September–October), while trends during the first rainy season (FRS, mid-May to mid-July) are insignificant. Here we analyze spatio-temporal variations in precipitation, aerosol, radiation, cloud and visibility observations from surface stations and from space to find indications for a potential contribution of anthropogenic air pollution to these rainfall trends. The proposed mechanism is that the dimming of incoming solar radiation by aerosol extinction contributes to reducing vertical instability and thus convective precipitation. To separate a potential aerosol influence from large-scale climatic drivers, a multi-linear regression model based on sea-surface temperature (SST) indices is used. During both LDS and SRS, weakly statistically significant but accelerating negative rainfall trends unrelated to known climatic factors are found. These are accompanied by a strong increase of pollution over the upstream tropical Atlantic caused by fire aerosol from Central Africa, particularly during the LDS. Over southern West Africa, no long-term aerosol records are available inhibiting a direct quantification of the local manmade effect. However, significant decreases in horizontal visibility and incoming surface solar radiation are strong indicators for an increasing aerosol burden, in line with the hypothesized pollution impact on rainfall. The radiation trend is further enhanced by an increase in low-level cloudiness. The large spatial extent of potentially aerosol-related trends during the LDS is consistent with the stronger monsoon flow and less wet deposition during this season. Negligible aerosol impacts during the FRS are likely due to the high degree of convective organization, which makes rainfall less sensitive to surface radiation. The overall coherent picture and the accelerating trends – some of which concealed by SST effects – should alarm policymakers in West Africa to prevent a further increase in air pollution, as this could endanger water supply, and food and energy production for a large and growing population.

# 1 Introduction

Sub-Saharan Africa in general, but particularly the already densely populated southern West Africa (SWA hereafter), currently experiences strong population growth and urbanization (United Nations, 2019). Together with economic growth in many sectors, this leads to an increasing demand for water. Currently, agricultural food production in SWA is mostly rain-fed with only basic or no irrigation systems in place (e.g., Namara & Sally, 2014). In addition, hydro-power is a crucial contribution to electricity production in many countries across SWA (e.g., the Volta Lake in Ghana, cf. Henley, 2019), which further increases demand. Together this creates a vulnerability to climate variability and long-term change, in particular with respect to rainfall (World Bank, 2012). Understanding the causes of rainfall variability and trends, particularly on interannual to decadal timescales, is crucial to make reliable predictions that allow the development of strategies for mitigation and adaptation.

The climate of SWA is strongly controlled by the seasonal evolution of the West African monsoon. The main dry season, when the convective zone lies further south, only lasts from December to February. The long wet period peaks during mid-May to mid-July (first rainy season, FRS) and in September–October (second rainy season, SRS), interrupted by the so called little dry season (LDS) (e.g., Thorncroft et al., 2011; Fink et al., 2017; Maranan et al., 2018). Meteorological conditions vary markedly between these four seasons and need to be taken into account to understand trends and variability. On interannual to decadal timescales, SWA is subject to marked rainfall variability, which has been linked to fluctuations in sea-surface temperatures (SSTs) in the nearby Atlantic Ocean with a moderate influence from the Pacific and Indian Oceans (Sutton & Hodson, 2005; Rowell, 2013; Diatta & Fink, 2014). The 1980s stand out as a particularly dry period but both satellite and station-based rainfall estimates show that SWA has undergone a mild recovery of rainfall since then, however with a large year-to-year variability (Sanogo et al., 2015). The contributions from the different seasons to this trend vary, with rainfall increases during the two rainy seasons (FRS and SRS) and a drying trend during the LDS (Sanogo et al., 2015; Nicholson et al., 2018a, b). While SST changes appear to have played a role in creating this trend (Diatta & Fink, 2014), the magnitude of seasonal changes, i.e., the trends during the FRS, LDS and SRS, remains poorly understood.

There has been recent speculations about a local influence of aerosol on rainfall in SWA (Knippertz et al., 2015b), an effect that has already been shown for southern Africa (Hodnebrog et al., 2016) and eastern China (Huang et al., 2016) for example. The basis for this is the dramatic increase of anthropogenic air pollution over recent decades (Liousse et al., 2014), the predominant source of aerosol in the region (Bauer et al., 2019). Aerosol particles can modulate rainfall through radiative (direct) and cloud (indirect) effects (Haywood & Boucher, 2000). Absorption and scattering reduces the amount of solar radiation penetrating to the surface (dimming), thereby increasing vertical stability and suppressing convective rainfall. Some aerosols act as cloud condensation or ice nuclei, thereby influencing cloud microphysics, albedo, and lifetime. The impact of this on precipitation is complex and depends on the cloud type and meteorological setting.

Ajoku et al. (2020) recently produced daily composites for August 2003–2015 based on aerosol optical depth (AOD) estimates from the Modern-Era Retrospective analysis for Research and Applications, Version 2 (MERRA-2) over the Gulf of Guinea (5–10°N, 10°W–10°E), which is affected by the advection of burning aerosol from mostly manmade agricultural and forest fires in Central Africa during this period (Giglio et al., 2006; Zuidema et al., 2016; Das et al., 2017). Comparing precip-

itation over SWA on clean and polluted days suggests a suppressing effect of the aerosol but it is difficult to establish a causal relationship due to strong covariance with meteorological variables, in particular low-level wind that changes from southerly on polluted days to easterly on clean days. One should therefore be cautious to link long-term aerosol changes with trends in precipitation based on these results. Unfortunately, there are no aerosol observations of sufficient quality over a long-enough time period over SWA itself, as satellite data suffer greatly from frequent cloud contamination (Hsu et al., 2012). This hinders establishing a more direct point-to-point relationship of aerosol with precipitation.

The recent Dynamics-Aerosol-Chemistry-Cloud Interactions in West Africa (DACCIWA) project (Knippertz et al., 2015a) conducted extensive field measurements in June-July 2016 (Flamant et al., 2018) accompanied by modeling experiments to better understand the role of aerosols in the West African monsoon system. DACCIWA studies confirmed the importance of an import of aerosol from fires in Central Africa in addition to local pollution sources (Menut et al., 2018; Haslett et al., 2019a). The strong monsoon flow during the LDS causes a fast spread of pollutants from the main sources in coastal cities inland (Deroubaix et al., 2019). Given the overall high concentration of aerosol particles and predominantly stratiform clouds, relatively little susceptibility of cloud microphysics to aerosol effects was found (Deetz et al., 2018b; Taylor et al., 2019). In contrast, the radiative effect appears to be significant, particularly as the very high relative humidity in the moist, deep monsoon layer leads to wet growth of aerosol particles (Deetz et al., 2018a; Haslett et al., 2019b). Combined with the high sensitivity of rainfall to destabilization by local radiative heating found by Kniffka et al. (2019), this creates potential for a significant direct effect. Consistently, sensitivity experiments have shown that changing anthropogenic emissions along the Guinea Coast has the potential to shift the entire monsoonal rainband (Menut et al., 2019) and regional circulation. However, climate models show substantial uncertainties in simulating the West African monsoon system (Roehrig et al., 2013; Hannak et al., 2017), casting doubt that realistic aerosol effects can be quantified with confidence using modeling approaches.

The goal of this paper is to provide new evidence that the documented strong increases of anthropogenic emissions in SWA and biomass burning aerosol imports from Central Africa have significantly affected decadal rainfall trends. Based on the recent DACCIWA results, we hypothesize that aerosol has a suppressing effect due to dimming, in particular during the LDS, when rainfall is mostly locally triggered (Maranan et al., 2018). For the FRS and SRS we expect a lesser import of aerosol from Central Africa (Giglio et al., 2006), less spreading of coastal pollution inland due to weaker monsoon winds, and more wet deposition due to the enhanced rainfall (see also the discussion of Fig. 3 in section 3.1). Given the lack of adequate aerosol observations and issues with numerical models as described above, we chose to concentrate solely on long-term observations of rainfall, radiation, and visibility from surface stations combined with selected satellite products. Seasonality, spatial distribution, and qualitative meteorological arguments will be used in the evaluation and interpretation of observed trends, while influences of other climatic factors such as SST variations are eliminated on the basis of a multilinear regression model. In section 2 we describe rainfall and other data sets used in this study together with a description of the techniques employed for the trend analysis. Section 3 contains a short discussion of the climatological background of our study region followed by a detailed analysis of rainfall and aerosol trends. We summarize the main results and draw conclusions in section 4. Results for the FRS, which is not in the focus of this paper, are provided for completeness in the Supplementary Material.

## 2 Data and methods

### 2.1 Region and season definition

For the definition of seasons, we follow the description already given in section 1: FRS (15 May–14 July), LDS (15 July–31 August), and SRS (1 September–31 October). Figure 1 shows a map of the region of interest in SWA. Longitudinally we will concentrate on the region 8°W–6°E, which avoids higher topographic features and contains all major cities along the Guinea Coast such as Abidjan, Accra, and Lagos as well as Kumasi in inland Ghana. Our main region of study reaches 3.5° of latitude inland from the coastline (bordered red in Figure 1), a distance for which we suppose to find aerosol effects on precipitation due to the fast northward transport of pollutants from the cities with the monsoon flow during the LDS. As during the SRS the monsoon flow weakens considerably, implying that pollutants will remain in the densely populated coastal plains, we concentrate on a smaller region, referred to as "coastal strip", during this season (bordered blue in Figure 1). The coastal strip only reaches 0.75° inland and is restricted to 8°W–4°E due to the strong curvature of the coast line to the east of Lagos.

### 2.2 Rainfall data sets and investigation period

Two different rainfall data sets are used in this study. First, the Climate Hazards Group InfraRed Precipitation with Stations (CHIRPS, Funk et al., 2015a) data set. It utilizes both high-resolution satellite imagery and in situ station data, and provides, amongst other things, daily rainfall estimates over land on a $0.25° \times 0.25°$ horizontal grid from 1981 onwards. CHIRPS uses an underlying static rainfall climatology (CHPclim, Funk et al., 2015b), which contains input from historical station precipitation averages, historical thermal infrared satellite estimation averages, and a global topographic grid. The analyses are performed every five days (pentad). First, the pentad estimates based on thermal infrared are corrected through local regression between cold cloud top duration and Tropical Rainfall Measuring Mission (TRMM) 3B42 precipitation (Huffman et al., 2007). In the second step, the product of the infrared estimates and CHPClim is bias-corrected using five-day accumulated gauge observations.The disaggregation to daily values is done using infra-red based cold cloud duration. Despite the product being relatively insensitive against changes in the station network (Chris Funk, personal communication) and a successful use in trend studies in SWA (e.g., Bichet & Diedhiou, 2018), large changes in surface station availability (e.g., https://data.chc.ucsb.edu/products/CHIRPS-2.0/ diagnostics/stations-perMonth-byCountry/pngs/Benin.003.station.count.CHIRPS-v2.0.png, last access: 8 May 2020) may induce inhomogeneities in long-term trends. This has been noticed by Diem et al. (2019) for western Uganda, who stress the necessity to validate satellite-derived trends by ground-based measurements. Furthermore, the algorithm tends to smooth spatial inhomogeneities.

Therefore as an additional source of precipitation data in the region, the Karlsruhe African Surface Station Database (KASS-D), containing daily, quality-controlled rain gauge measurements from manned weather stations operated by National Weather Services (e.g., Vogel et al., 2018) has been used in this study. Only stations with at least 50% data coverage are considered to allow a meaningful trend analysis. As this criterion is hardly fulfilled before 1983 and after 2015, the main analysis is restricted to this period. In addition, the trend analysis is repeated for the more recent, shorter timespan 2001–2017, for which surface observations of incoming solar radiation and more satellite data are available (see section 2.4). Unfortunately, the availability

of KASS-D data deteriorates during this period, mostly in Nigeria where our data base has little data after 2015. Despite this, we decided not to end this recent period in 2015 because trends become less meaningful for shorter timespans. Note that in the 1990s to 2010th, KASS-D contains daily data from many stations in SWA that have not been used in CHIRPS.

## 2.3 Other surface, satellite-based and reanalysis data sets

Visibility and low and medium cloud cover data from the Met Office Integrated Data Archive System (MIDAS) (Met Office, 2006) are used from those stations in SWA where data are available for at least 50% of the days in each season. Horizontal visibility data are categorized into ranges of "below 10 km", "10–20 km", and "above 20 km" for which time series and trends are calculated. Observations of surface downwelling shortwave radiation (SDSR) are available at Parakou (October 2001–June 2017) and Lamto (January 2001–May 2018). The instrument in Parakou was replaced in March 2009 and again in March 2014, potentially influencing trend calculations due to inconsistencies in the measurements. However, a shorter time series (2002–2015) from Djougou, about 100 km northeast of Parakou, indicates that the observations in Parakou are consistent (not shown). No visibility data are available for Lamto, but human observer estimates of total cloud area fraction (TCAF) for January 2000–July 2016. From these (sub-)daily observations seasonal, respectively monthly, averages were calculated, again with a 50% data coverage criterion. For a more complete look at radiation, monthly data of SDSR and effective cloud albedo (ECA) of the SARAH-2 data set from the Satellite Application Facility on Climate Monitoring (CM SAF, Pfeifroth et al., 2017) are analyzed. The comparison of normalized trends (see section 2.4) of SDSR and ECA allows us to estimate a residual potentially related to aerosol. This technique of normalization is also applied to observations from the surface stations in Parakou and Lamto.

Monthly satellite measurements of aerosol optical depth (AOD) on a $1° \times 1°$ horizontal grid from the Moderate Resolution Imaging Spectroradiometer (MODIS) (Platnick et al., 2017), i.e., the "combined dark target and deep blue AOD at 0.55 micron for land and ocean: mean of daily mean", are used to calculate seasonal trends between July 2002 (beginning of data set) and October 2018. As these are monthly data, they are weighted when calculating season-averaged values. AOD for the LDS (15 July–31 August), for instance, is computed as AOD(LDS)=(0.5·AOD(July)+1.0·AOD(August))/1.5. Especially over land clouds often inhibit AOD measurements from space, leading to many missing values in the MODIS monthly products. To cover the diurnal cycle, only months are used, when AOD data from both the Aqua and Terra platforms are available, which are then averaged to obtain one single monthly value. For every year a seasonal mean value is computed if data are available for all months of the respective season. Again a sufficiently complete record with data for at least 50% of the seasons between 2002 and 2018 is required before calculating a trend.

To give climatological context to the trend analysis presented here, relative humidity, cloud cover, and meridional wind speed data from the fifth generation of European Centre for Medium-Range Weather Forecasts (ECMWF) atmospheric reanalyses of the global climate (ERA5, Hersbach et al., 2020) are used. The influence of ocean temperatures on rainfall is analyzed using the five most important SST-based climate indices for the region, namely Atlantic 3 (3°S–3°N, 0–20°W), Niño3.4 (5°S–5°N, 120–170°W), the coupled ocean-atmosphere Atlantic Meridional Mode, Indian Ocean (10°S–30°N, 50–90°E) and the Atlantic Multidecadal Oscillation index (Atlantic Ocean from 0–70°N), all as used by Diatta & Fink (2014). The Atlantic Meridional

Mode is based on the National Centers for Atmospheric Prediction/National Center for Atmospheric Research (NCEP/NCAR) reanalysis and the Atlantic Multidecadal Oscillation index on the Kaplan SST data set. The other three indices are computed from SST values from the Hadley Centre Sea Ice and Sea Surface Temperature (HadISST) data set. The same weighting as for the AOD data is used for the conversion from monthly climate indices to seasonal averages.

## 2.4 Methods for trend analysis

Long-term trends for seasonal rainfall totals are computed for every CHIRPS grid point and KASS-D station, applying Sen's slope method (Sen, 1968; Hirsch et al., 1982; Hipel & McLeod, 1994). It calculates slopes for each pair of two consecutive years in a time series and the final trend is the median from all slopes. These trends are tested on statistical significance using the Mann-Kendall test (Mann, 1945; Davison & Hinkley, 1997; Hipel & McLeod, 2005). Generally, trends are considered statistically different from zero if the two-sided $p$-value is smaller than the tested significance level $\alpha$. All our tests are performed for $\alpha$-values of 5% and 20%, the latter being a relatively weak criterion for statistical significance. As discussed in the context of "climate change attribution" by Lloyd & Oreskes (2018) and Knutson et al. (2019), the choice of significance levels depends on how one intends to interpret the results. Small $\alpha$-values reveal high confidence that a trend found in the data has some other reason than natural variability. However, a less strict significance level ($\alpha = 20\%$ in our case) can still be useful for a study like ours that is dealing with the challenge of a relatively short and incomplete data record subject to multiple influence factors and possibly non-linearities. In such a situation choosing a large $\alpha$ implies reducing so-called "type II errors", i.e., retaining the null hypothesis of no trend beyond natural variability, although such a trend actually exists but is hard to detect with the information at hand. Taking into account additional factors such as geographical distribution and seasonal behavior can help in the evaluation of a trend with weak statistical significance. Ultimately, it is the balance of all available evidence that does or does not suggest that an identified trend has other than natural causes (Knutson et al., 2019) and this is the philosophy we are following in this study. We feel that such an approach is particularly justified in the given situation, as a risk assessment for a potential local human influence on rainfall in SWA is urgently needed.

Interannual variations of precipitation in SWA are, dependent on the exact region and season, influenced by (remote) climate indices. In order to distinguish between these climatic signals and other parameters such as (local) air pollution, we calculate so-called "residual" time series through subtraction of a linear regression model based on five climate indices defined in the previous section from the original time series. For each time series, 31 different linear models are constructed that consist of either single indices (i.e., five linear models) or any combination of the five indices (i.e., ten combinations with two indices, ten combinations with three indices, five combinations with four indices, and the one linear model with all five indices). For each model the leave-one-out cross validation procedure (Efron & Tibshirani, 1997, for example) is performed and evaluated calculating the residual sum of squares. Aiming to keep the model as simple as possible, the linear regression was constructed to consist of as few indices as possible to avoid (near) multicollinearity. Only if a combination of two indices yields a model for which the residual sum of squares is at least 3% lower than that of a single-index linear model, the two-index model is chosen. A third, fourth, or fifth index is again only added, if the performance of such a linear model increases by more than 3%

compared to a model consisting of less indices. Thresholds between 1% and 5% were tested for the performance improvement but all result in the same choice for the linear model at the majority of CHIRPS grid points and stations.

The optimal linear model determined this way and its correlation coefficient with the original rainfall time series are shown in Fig. 2 for the LDS and SRS during the 1983–2015 period. During the LDS a simple linear model built on the Atlantic 3 index alone yields best results for the large majority of CHIRPS grid points analyzed (yellow in Fig. 2a). This is consistent with previous results for a slightly larger region along the Guinea Coast and a longer season from June to September (Diatta & Fink, 2014). Exceptions are: (a) the Kwahu Plateau in southwestern Ghana (combination of Atlantic 3 with the coupled ocean-atmosphere Atlantic Meridional Mode, green), (b) central Ghana and the far southwest of Ivory Coast (combination of all five climate indices, pink), and (c) northern Benin and adjacent Nigeria (Atlantic Multidecadal Oscillation index, orange), but this region consists of only five grid points. The agreement between CHIRPS and the stations is largely good, with the exceptions of Kara in Togo (Atlantic Multidecadal Oscillation index, orange), Sokodé in Togo (Niño3.4 index, turquoise), and Warri in Nigeria (Indian Ocean, blue). We are not aware of any convincing physical reasons for these deviations and therefore mostly consider them as statistical fluctuations. This is consistent with Fig. 2b showing correlations with the best models to be above 0.5 through most of the region, largely irrespective of model choice. Only the orange points from Fig. 2a at the far northeastern fringe of the study region show a lower correlation, indicating problems with data or stronger terrestrial forcings farther away from the ocean. The simple picture that emerges from this analysis is that a considerable part of the season-to-season variability is linearly controlled by the SST over the nearby Atlantic Ocean, with warmer waters leading to more rainfall. This is related to a weaker temperature and pressure gradient towards the Sahel, which leads to stronger convergence and higher moisture content in SWA (Losada et al., 2010).

The picture during the SRS is also dominated by the Atlantic 3 index but in the northern and especially eastern parts of the region, other climate indices yield better linear models (Fig. 2c). Given the weaker monsoon flow during this season, this may indicate a less close link to the nearby Atlantic Ocean and more room for teleconnections to influence rainfall. Correlation coefficients are generally lower than during the LDS but still reach 0.5 across considerable parts of the region, particularly in the coastal strip (marked by a black line in Fig. 2d). It is striking that the regions that do not show the simplest model based on the Atlantic 3 index also show lowest correlations, which again points towards data problems and/or more local influences such as topographic forcing by the Oshogbo Hills in Nigeria.

During the FRS, the Atlantic 3 index is also dominating but regions where other indices yield better linear models are larger than during the other two seasons (Supplementary Fig. 1a). The correlation of the best models with rainfall is low during the FRS rarely exceeding values of 0.5 (Supplementary Fig. 1b) showing a rather weak effect of ocean temperatures on rainfall during this season. As before, the agreement between station and CHIRPS data in both the optimum index combination and the correlation is very good.

The optimal linear models described above are now used to remove the dominant influences of SST fluctuations from the original time series. Trends in these residual time series will differ from those computed from the original time series and we will refer to them as "residual" trends and "full" trends, respectively. Trends and their statistical significance are also calculated

for all other observed parameters described in section 2.3, namely visibility, clouds, radiation, and AOD, but no residual time series are calculated in these cases.

## 2.5 Indirect indicators for aerosol trends

Given the issues with aerosol observations in SWA described above we have to rely on indirect indicators to determine changes in air pollution. These are observations of cloud cover, radiation, and horizontal visibility. As there is not sufficient information for even the simplest radiative transfer estimates, we chose to estimate a potential aerosol effect on radiation through considering normalized trends. In this context, normalizing means subtracting the mean and dividing by the standard deviation of the respective time series. The dimensionless trends in such normalized time series can be directly compared to each other assuming no significant changes in variance during the considered time period (stationarity). If surface radiation was entirely determined by cloud cover, identical but opposite normalized trends would result. A non-zero sum of the normalized trends points to an additional aerosol effect and / or changes in cloud optical thickness, which could at least in parts be related to aerosol. Positive sums (e.g., slight reduction in radiation but strong increase in cloud) would then indicate reduced aerosol, while negative sums (e.g., strong reduction in radiation but only slight increase in cloud) points to increased aerosol. This concept will be applied to the time series of CM SAF satellite data and surface observations from Parakou and Lamto in section 3.3.

Another data set we use as indirect aerosol indicator are surface observations of horizontal visibility as estimated by human observers. Trends in this parameter are most likely an effect of changes in the aerosol burden but changes in low-level humidity and clouds may also have an effect, as they change incoming light and the split between direct and diffuse radiation.

## 3 Results

In this section, trends in rainfall (section 3.2) and various indicators of aerosol burden (section 3.3.) are presented and related to each other. To put the results into context, the short section 3.1 summarizes the annual cycle of relevant meteorological variables.

## 3.1 Climatological conditions

Figure 3 shows the time mean seasonal evolution of key variables based on ERA5 and CHIRPS rainfall averaged over the main study region shown in Fig. 1. The main seasons as used in this paper are delineated by vertical black lines. During the main dry season from December to February rainfall drops to values well below 1 mm day$^{-1}$ followed by a gradual ramp-up towards the FRS starting in mid-May. Peaks during the two rainy seasons FRS and SRS reach about 8 mm day$^{-1}$, while rainfall drops to below 4 mm day$^{-1}$ during the LDS. The monsoonal retreat in October and November occurs much more rapidly than the onset.

Maranan et al. (2018) recently developed a seasonal climatology of rainfall types over the region based on satellite imagery and re-analysis data. They found that the location of the African easterly jet close to the Guinea Coast during the FRS creates

enhanced vertical wind shear, leading to the highest degree of convective organization into long-lived mesoscale clusters during the year (marked with ++ in Fig. 3). In contrast, during the SRS, the vertical profiles of shear and instability lead to a more local convective triggering, for example by the land-see breeze, and therefore smaller and shorter-lived but often intense systems (+ in Fig. 3). Conditions during the LDS are significantly different. During this period, SWA is subject to a strong southerly monsoon flow, leading to fast advection of thermodynamic and chemical properties from the Gulf of Guinea, typically along a low-level jet that forms at night. Rainfall during the LDS is often due to relatively weak, isolated, short-lived, and weakly organized convection (∘ in Fig. 3 Maranan et al., 2018), mostly occurring in the afternoon. Land-sea breeze days have a seasonal minimum (Guedje et al., 2019) and stability is often too high to allow the transition from cumulus clouds in the coastal hinterland into deep cold convective clouds in the course of the afternoon.

The arrows in Fig. 3 show average wind profiles for the three seasons of interest. Clearly the monsoon flow is strongest in the LDS reaching meridional wind speeds exceeding 4.5 m s$^{-1}$ at 950 hPa with southerly winds up to 875 hPa, marking the depth of the monsoon layer. The FRS is also characterized by marked southerlies and an even deeper monsoon layer, while the SRS shows weaker flow reduced to a shallower layer. This is consistent with Guedje et al. (2019), who investigated upper air data at the coastal station of Cotonou. As already discussed earlier, we anticipate that this should restrict the largest aerosol effects to the coastal strip, which contains the main pollution sources. The higher rainfall in SRS as compared to the LDS should also lead to more wet deposition, which would further support the concentration to the vicinity of sources. Wet deposition will of course also be enhanced during the FRS.

Another parameter of interest is the frequency of relative humidity of 95% and higher (light blue curves in Fig. 3), as this determines the potential for wet growth of aerosol particles, which strongly enhances radiative effects (Deetz et al., 2018a; Haslett et al., 2019b). Here large differences are seen between the deep layer of very moist conditions in LDS as compared to the much shallower layers in the rainy seasons with maxima at 950 hPa (FRS) and even 975 hPa (SRS). Qualitatively, these results suggest that during the LDS the potential for wet aerosol growth is higher and that considerable aerosol effects could in fact be spread over a much larger region through the combination of faster transport and less wet removal than in the other seasons. For the LDS and the SRS, time series of the frequency of relative humidity of 95% and higher (Fig. 4) reveal slight decreases over 1983–2015 of about 0.1% yr$^{-1}$, which are significant only on the $\alpha = 20\%$ level. Given the highly non-linear dependence of wet growth of aerosol particles on relative humidity (e.g. Haslett et al., 2019b), this relatively small change implies that the radiative effect per amount of dry aerosol will be smaller in the later part of the study period. As our estimates for aerosol trends rely on horizontal visibility and surface radiation, the change in dry aerosol may be even larger than deduced from these parameters alone. The trend of relative humidity exceeding 95% during the FRS is indistinguishable from zero (Supplementary Fig. S2 top).

Finally the red curve in Fig. 3 shows the mean vertical profile of cloud cover at 06 UTC (corresponds to local time in the study region), the analysis time closest to the diurnal maximum of low clouds (van der Linden et al., 2015). These clouds are crucial for the radiative energy budget and, thus, for vertical mixing and the triggering of convection in the afternoon (Kalthoff et al., 2018; Kniffka et al., 2019). We anticipate this factor to influence the significance of aerosol changes to the surface

energy balance, when incoming radiation is strongly reduced by cloud cover. Peak coverage at about 950 hPa differs only little between the three seasons, but in the LDS the cloud layer is somewhat thicker.

For the remainder of the paper, the analysis will concentrate on the LDS and the SRS, for which we found the largest indications for aerosol effects. The corresponding results for the FRS are provided in the Supplementary Material and will be referred to along the way. Figure 3 offers relatively little in terms of potential reasons of fundamentally different behavior in the FRS, so we hypothesize that the much larger degree of convective organization during this season found by Maranan et al. (2018) is key in reducing sensitivity to local aerosol effects. In addition, spatial patterns of positive and negative rainfall trends during FRS change depending on the time period analyzed (cf. Supplementary Fig. S3a, b with S3d, e) and there are more pronounced discrepancies between CHIRPS and station data. Regionally averaged trends are weak and statistically not significant on the 20% level (Supplementary Fig. S3c, f).

## 3.2  Rainfall trends

### 3.2.1  Little Dry Season

Figure 5 shows full (top row) and residual (i.e., with the influence of SST variations removed, see section 2.4; middle row) rainfall trends for the LDS during 1983–2015 (left) and 2001–2017 (right) for both surface station observations and CHIRPS. The full trend over the long period 1983–2015 (Fig. 5a) largely confirms results by Sanogo et al. (2015) of an overall drying in the region during the LDS. Largest absolute decreases on the order of $-0.04$ mm day$^{-1}$ yr$^{-1}$ are found over the wet Niger Delta region in the southeastern corner of the study region and in the far west along the border of Ivory Coast and Liberia. Large parts of Ivory Coast and Ghana show moderate drying, while trends over the eastern countries (Togo, Benin, Nigeria) are somewhat more mixed. Weakly positive trends are seen in the coastal plains between Accra and Lagos and in the far north, in the surroundings of the Atakora Mountains and the Oshogbo Hills. A remarkable pattern evident from Fig. 5a is that, while inland the agreement between stations and CHIRPS is largely good, many stations at the immediate coast show positive trends, some quite considerable ones, which are not always reflected well in CHIRPS. This points either to data problems in CHIRPS at the coast or to a very local effect, as for example related to the sea breeze (Guedje et al., 2019). There are indications that the ongoing deforestation and urbanization in the coastal strip enhances convective triggering during the afternoon due to increased turbulent fluxes of sensible heat (Chris Taylor, personal communication).

Despite the relatively high correlations with SSTs over the Atlantic (see Fig. 2), the residual trends (Fig. 5b) do not differ fundamentally from the full trends. The main reason for this is the small warming of the tropical Atlantic during this season (Fig. 5c) that dominates the statistical model at the majority of grid points and stations (see section 2.4). Amongst the few exceptions are Central Ghana and western Ivory Coast that both show a combination of all five climate indices to give the best statistical model (Fig. 2a). The Niño3.4 index, for example, is anti-correlated with rainfall and increases by 0.1 K per decade (not shown). This explains that residual trends are more positive than the full trends in regions where this index is part of the best linear model. Nevertheless, overall the highly negative residual trend points to a possible effect of increased dimming by aerosol. This would not stand in contrast with the positive trends at coastal stations as also seen in Fig. 5b, because the

additional aerosol radiative forcing needs time to take effect over land, while the immediate coast is dominated by advection from the ocean (and was most strongly changed by urbanization). The positive trends to the north of the Kwahu Plateau in Central Ghana, the Atakora Mountains in Togo and Benin, and the Oshogbo Hills in northwest Nigeria could be an indication of delayed convective triggering over higher ground due to the reduced solar insolation, allows rainfall systems to travel further downstream.

Figure 5c shows the CHIRPS trends averaged over all grid points of the study region. Not surprisingly, both full and residual time series are quite similar with relatively small trends as compared to the large interannual variability. Subtraction of the SST-based statistical model reduces the trend from $-0.026$ mm day$^{-1}$ yr$^{-1}$ to $-0.022$ mm day$^{-1}$ yr$^{-1}$ but, since the interannual variability is also reduced, this leads to statistical significance on the 20% level. The latter value translates to 23 mm per month over the 33-year period, which corresponds to nearly 20% of monthly rainfall during the LDS.

The right-hand side panels in Fig. 5 show the corresponding analysis for the shorter and more recent period 2001–2017. As for the longer period 1983–2015, there is a general tendency for inland areas to dry, while coastal areas even get wetter, but this contrast is now much sharper. Removing the contribution from an overall positive SST trend over the tropical Atlantic of 0.1 K per decade (Fig. 5f) has a much larger effect than for the longer period, with trends becoming more negative practically everywhere. Positive values are now restricted to the densely populated strip from Accra to Lagos, the latter by far the largest city of the region and subject to fast expansion. The area-averaged trend of the full time series now amounts to $-0.039$ mm day$^{-1}$ yr$^{-1}$ while the residual trend reaches $-0.072$ mm day$^{-1}$ yr$^{-1}$, which is 2.4 times larger than for 1983–2015 and statistically significant on the 20% level. This reduces typical monthly rainfall amounts from 150 mm to 112 mm, i.e., by one quarter, indicating that the effect we are postulating here is accelerating with the fast growing population.

### 3.2.2 Second Rainy Season

The corresponding analysis for the SRS is shown in Fig. 6. The full long-term trend (Fig. 6a) is characterized by drying in southern Ivory Coast and southern Ghana, moderate moistening in central Ivory Coast and central Ghana, as well as strongly positiv trends in the northeast. Averaged over our study region, the trend is positive, which is consistent with recent studies by Bichet & Diedhiou (2018) and Nkrumah et al. (2019). As for the LDS, the overall agreement between stations and CHIRPS is good but coastal stations tend to show more positive trends than the immediate CHIRPS neighbors.

The dominating positive influence of the Atlantic SSTs on rainfall (Fig. 2c and d) combined with a positive trend during the SRS (Fig. 6c) leads to an overall more negative precipitation trend after removal of the SST-based statistical model (Fig. 6b). Throughout the coastal strip (marked by a pink line in Fig. 6a and 6b) rainfall trends are now decreasing almost everywhere but negative values stretch much farther into central Ivory Coast and Ghana, while the positive values over the Atakora Mountains and the Oshogbo Hills are retained. This pattern reflects the regions where anthropogenic aerosol is assumed to have a major effect (close to the coast) and a minor or no effect, respectively (far inland and over hilly terrain) for the following reasons: As discussed in section 3.1, inland transport are much weaker during the SRS than during the LDS (Fig. 3) and wet deposition should be larger. Therefore it is to be expected that any aerosol influence is more confined to the vicinity of the major urban centers along the coast. The negative impact continues over flatter and drier inland areas, while hilly regions may even receive

more rainfall, since they stand out above the polluted boundary layer, and instability and water are not so frequently removed elsewhere. Positive trends in the northern parts of the region during the SRS are consistent with a projected increase of rainfall in the Sahel associated with a delayed withdrawal of the WAM (Monerie et al., 2016).

Averaging CHIRPS trends across the coastal strip leads to an insignificantly small trend in the full time series of $-0.010$ mm day$^{-1}$ yr$^{-1}$ (Fig. 6c) given the contrast between western and eastern regions (Fig. 6a). The residual trend, however, reaches

$-0.036$ mm day$^{-1}$ yr$^{-1}$ and is thus statistical significant on the 20% level. As for the LDS, this corresponds to a reduction in monthly rainfall of about 20% (or 35 mm) over the 33-year period.

Repeating the analysis for the shorter and more recent period from 2001–2017 (right panels in Fig. 6) does again lead to a considerable shift to more negative trends, which supports the idea of an accelerating human influence. Note, however, that for the more recent period, also the full trend becomes significantly negative over large parts of the study region when compared

to the last three decades. Particularly after computing the residual (i.e., after removing the impact of warming SSTs during this period), areas with positive trends become restricted to central Ivory Coast, the northern Lake Volta, and smaller regions in Nigeria, while large parts of SWA show negative trends as low as $-0.04$ mm day$^{-1}$ yr$^{-1}$ (Fig. 6e). The change is most dramatic east of Lake Volta in Togo and Benin. Most of the large coastal cities, with the exception of eastern Ghana, now also show clearly negative residual trends.

Spatial averages over the coastal strip (Fig. 6f) show a corresponding shift to more negative trends. For the full time series this amounts to $-0.050$ mm day$^{-1}$ yr$^{-1}$, while the residual trend is $-0.087$ mm day$^{-1}$ yr$^{-1}$ and thus 3.3 times larger than for 1983–2015. The residual trend is statistically significant on the 20% level, partly also due to a reduced year-to-year variability when SST effects are removed. Over the 17-year period, this trend corresponds to 45 mm per month or 90 mm over the September-October period of the SRS. This is a substantial reduction relative to typical SRS totals of 350 mm and annual

totals of 1400 mm (Sanogo et al., 2015).

### 3.3   Indirect indicators for aerosol trends

In the previous section, we have demonstrated that – once influences of SST changes are corrected for – large parts of SWA have undergone an accelerating drying over recent decades. Seasonal and geographical patterns, together with the acceleration, are consistent with the hypothesis of a human influence through rapidly growing emissions of pollutants. What other evidence

do we have to support this idea? As we will discuss in more detail in the next subsection, usable aerosol measurements are largely restricted to the ocean adjacent to SWA, impeding the establishment of a direct link to rainfall. Therefore we turn here to indirect indicators such as horizontal visibility (section 3.3.2) and SDSR (section 3.3.3), which need to be regarded in concert with cloud cover that influences both quantities.

### 3.3.1   Satellite-based aerosol estimates

Figure 7 shows trends in AOD as estimated from MODIS over the period 2002–2018, which is almost coincident with the time span used for the rainfall trend analysis in section 3.2. The analysis was restricted to pixels with a sufficiently complete record (see section 2.3 for details), which leaves us with practically no useful information over SWA during the LDS and at best very

limited data during the SRS. Such problems are also present for other satellite sensors and are related to the very frequent cloud cover (Hsu et al., 2012). The following analysis therefore focuses on MODIS AOD over the Gulf of Guinea from where aerosol particles are frequently transported into the study region with the predominantly southerly flow (Deroubaix et al., 2018).

During the LDS (Fig. 7a), MODIS AOD data show a spatially consistent increase over the northern hemispheric tropical Atlantic out to 10°W and encroaching into the coastal areas. Maximum increases are found in the northeastern parts where AODs increase by more than 0.1 during this period (maximum 0.13). Given typical mean values of about 0.4 near the Guinea Coast (not shown), this value underlines the dramatic increase in pollution over this large area. This phenomenon has been described extensively in the literature and is related to the increase in biomass burning aerosol from Central Africa during the local dry season (Mari et al., 2011; Andela et al., 2014). A number of recent field campaigns in the southeast Atlantic targeted this aerosol layer and its interaction with clouds specifically (Formenti et al., 2019, and references therein). The fire plume typically gets advected westward to the equatorial Atlantic, from where dry and cloud-related vertical mixing inject aerosol into the monsoon layer. Southerly or southwesterly winds then carry it northward toward SWA (Dajuma et al., 2020). Model experiments and aircraft measurements in the framework of DACCIWA have shown a considerable contribution to air pollution along the Guinea Coast in addition to the rapidly increasing local emissions (Menut et al., 2018; Haslett et al., 2019a). Ajoku et al. (2020) related daily variations of this aerosol plume during August to daily variations in rainfall over SWA, and found a clear negative impact, most significantly over the eastern part of our study domain, where the AOD trends are largest. However, as already discussed in the introduction, one has to be cautious to claim a direct causality here, as the daily AOD variations are associated with significant changes in circulation. Nevertheless, it appears plausible that the dramatic increase of pollution import from Central Africa has contributed to the rainfall trends discussed in section 3.2.1.

As expected, AOD trends over the tropical Atlantic during the SRS (Fig. 7b) and the FRS (Supplementary Fig. S4) are markedly smaller than during the LDS. Typical values for the SRS vary between 0.04 per decade in the eastern parts to 0.02 in the west. This is mostly related to the fire zone shifting further into the southern hemisphere (Mallet et al., 2019) with the onset of the rainy season in the southern hemisphere inner tropics. An additional factor may be the slower transport with the much weaker monsoon winds as evident from Fig. 3. Over the entire 17-year period, the increases shown in Fig. 7b still amount to around 0.05, respectively, which is considerable given the now reduced background values of about 0.26 (not shown). Assuming no large changes in anthropogenic emission in SWA between the LDS and SRS should therefore shift the relative importance to local sources. A somewhat surprising result are the negative trends at the far northern fringes of the domain, i.e., mostly between 9 and 10°N. These are marginally statistically significant but there is at least some east-west consistency to this signal. Given the southward shift of the rainbelt between the LDS and SRS, this area is now coming to the end of its local rainy season and the decrease in cloudiness allows more frequent aerosol measurements from space. The reasons for the reduced aerosol are not clear at the moment. Assuming an increased burden over coastal areas of SWA, possible candidates are an increase in wet deposition during northward transport and a change in circulation and thus transport. The former likely plays a role in the far northwest and downstream of Lake Volta, where rainfalls are in fact increasing (see Fig. 6b).

### 3.3.2 Station-based visibility estimates

Horizontal visibility is regularly estimated by human observers at standard weather stations. Requesting a certain level of data completeness leaves 12 stations across SWA to investigate for the period 1983–2015, also used for the long-term rainfall trend analysis in section 3.2. Averaged over all stations a dramatic increase in the lowest range "below 10 km" at the cost of the two other ranges "10–20 km" and "above 20 km" is evident in all three seasons (LDS and SRS in Fig. 8, FRS in Supplementary Fig. S2 bottom). All trends are significant on the $\alpha = 5\%$ level and reach about 1.2% yr$^{-1}$ in the "below 10 km"-category. Visibilities of more than 20 km were already rare in the 1980s and 1990s and do almost vanish after the year 2000.

Figure 9 shows corresponding trends for the individual stations. The frequency in the "below 10 km"-category increased significantly at most stations in all three seasons (LDS and SRS in Fig. 9, FRS in Supplementary Fig. S5). The largest increase of about 3% yr$^{-1}$ is found in Gagnoa in Ivory Coast during the LDS (Fig. 9a) and the SRS (Fig. 9b). Large reductions in visibility are also found in Savè, Bohicon, and Cotonou. The latter is located at the coast from where pollutants are transported northward with the main flow towards Savè and Bohicon. For the other coastal cities the increase in the lowest category is largest in Abidjan, especially during the SRS (Fig. 9b) followed by Lomè and Accra. In Accra the decrease in horizontal visibility becomes apparent mainly in the decrease of the frequency in the "above 20 km"-category of about 2.5% yr$^{-1}$ in both seasons. Roughly since the year 2000 there are practically no observations at any station in this category, as the mean over all stations reveals (Fig. 8). More moderate decreases in visibility are found for the Ivorian stations Yamoussoukro, Dimbokro, and Daloa, which may be related to the predominantly southwesterly transport away from the main urban centers Abidjan, Accra, Lomé, Cotonou, and Lagos (Deroubaix et al., 2019). This pattern, however, is not inconsistent with an aerosol impact on precipitation trends (see section 3.2), as rainfall systems generally tend to travel westward. A special behavior is found for Atakpame, where the frequency in the "10–20 km"-category increases at the cost of both other categories, meaning that both the very good and very bad conditions become less frequent. These trends, however, are mostly not or only marginally significant. Trends at the northernmost and elevated station Parakou in central Benin show the strongest increase in visibility of all stations. Observations in the lowest category decrease while frequencies in both other categories increase during the LDS (not significantly for "above 20 km") and in the "10–20 km"-category only during the SRS, respectively. The reason for this is unclear but could be related to a reduction in local pollution sources or changes in observation procedure. The former appears rather unlikely, as political efforts to reduce local emissions are still in its infancy across the region and also given the overall population increase. The latter is not immediately obvious from the data record and not documented to the best of our knowledge. In any case, the stark contrast to upstream Bohicon and Savè casts some doubt on the homogeneity of the data. Reducing the considered time period for Parakou to 2001–2017 (for which radiation data are also available, see Fig. 12) yields more neutral or even negative trends in visibility (not shown). The positive trend during the SRS could be related to increased rainfall and thus wet deposition, as already discussed for MODIS AOD in the previous section.

It is generally conceivable that changes in low cloud cover could have influenced the visibility estimates in addition to changes in aerosol. Respective trends are shown in Fig. 10 for the 12 stations used in the visibility analysis plus Lamto in Ivory Coast (only since 2000). The cloud trends are overall not as systematic as for visibility. There is no obvious covariance between

the two trends that would suggest a systematic influence. In Accra, for example, cloud cover strongly decreased throughout the three seasons, while at the same time horizontal visibility is strongly reduced. Nearby Lomé in contrast shows a reduction in visibility but increase in clouds. As discussed in section 3.2, the coastal stations show an abnormality in rainfall trends, which may point to changes in the land-sea breeze circulation. This would naturally also affect visibility and cloud cover. Overall, the combined visibility and cloud analysis supports the hypothesis of increased aerosol loadings in coastal and low-lying inland areas and their potential role in rainfall reduction.

### 3.3.3 Station- and satellite-based radiation estimates

An even stronger indicator for aerosol dimming than horizontal visibility is SDSR. Trends in this quantity will generally be a combination of changes in aerosol and clouds, making it more challenging to analyze quantitatively. As explained in section 2.5, the strategy is to rely mostly on normalized trends and to deduce a possible aerosol effect indirectly.

Time series for the LDS and the SRS of the CM SAF satellite data for radiation (SDSR) and cloud cover (ECA) in Fig. 11 and for the FRS in Supplementary Fig. S2 (middle) show clear anti-correlation and opposite long-term trends. These are significant on the $\alpha = 5\%$ level during the FRS and for SDSR during the SRS when the trend of ECA is significant on the $\alpha = 20\%$ level. The two LDS time series reveal strong interannual variations and therefore trends are not significant on the tested $\alpha$-levels (Fig. 11). The trends in SDSR are consistent with the marked decreases in horizontal visibility discussed above and further support the idea of an increase in aerosol burden.

Figure 12 shows a monthly distribution of absolute and normalized trends in radiation and clouds for selected surface stations and from satellite. As the only two stations with long enough records, Lamto (2001–2018) and Parakou (2001–2017) are both far away from the coastal cities, we expect them to be representative for inland conditions not affected directly by the sea breeze. Despite the large distance between them, both stations show a consistent reduction in SDSR during all months with most significant changes of about $-2$ W m$^{-2}$ yr$^{-1}$ restricted to the LDS and SRS (Fig. 12a). Values remain lower at Lamto during November and December, when northern Parakou is already entering the dry season. Accumulated over the period of available data leads to a maximum monthly reduction of almost 20 W m$^{-2}$ yr$^{-1}$, which is considerable given regional background values of about 170 W m$^{-2}$ during boreal summer (Knippertz et al., 2011; Hannak et al., 2017; Hill et al., 2018). Both Lamto and Parakou saw a significant increase in cloudiness during the LDS and SRS (Fig. 10) and these likely explain part of the trend.

To gauge the relative importance of aerosol and cloud changes, Figs. 12b and 12c show normalized trends of SDSR and LLC for Parakou and Lamto, respectively. For Parakou, this analysis results in a smoothing of the annual cycle of the negative SDSR trends but the minimum during the LDS and SRS remains (blue line in Fig. 12b). Cloud changes are near zero from January to May and then remain positive but small for the rest of the year (red line in Fig. 12b). Making the assumptions outlined in section 5, this implies a residual trend that we assume to be due to aerosol during the month from June to October with a peak in September (golden line in Fig. 12b). The corresponding analysis for Lamto (Fig. 12c) shows consistent behavior for SDSR, i.e., an overall smoother evolution after normalization with a minimum during the LDS and SRS, but a much larger positive trend in cloudiness. The residual – supposedly aerosol-related – trend is near zero from June to November and even

negative in the rest of the year with a minimum in April. A caveat in this result is that Lamto does not provide the standard
cloud cover reports available for Parakou but a more qualitative estimate called total cloud area fraction (TCAF), which maybe
less reliable, particularly when human observers change over time. If however we deem the results to be reliable, they would
suggest that the wet season effect we see at Parakou and in most visibility estimates is weak in Lamto, for which we do not
have any visibility data to back up the radiation analysis. Being a remote forest observatory and located relatively far west, so
potentially not much affected by northeastward advection from the big coastal cities and biomass burning aerosol from Central
Africa, a small aerosol signal is plausible. Nevertheless, the signal during the rest of the year remains surprising and may point
to an increase in rainfall during the Sahelian dry season and accompanied washout of aerosol particles (Sanogo et al., 2015).
As this part of the year is not in the focus of this study, we leave a more detailed analysis to future work.

Finally, Fig. 12d shows a corresponding analysis entirely based on satellite data and averaged over the study region (see
Fig. 1 for area definition). The period covered here spans 1983–2015, as for the long-term rainfall trends discussed in section
3.2. Satellite-based, normalized SDSR mostly shows negative trends but less smooth than for the two stations. Values remain
negative (and mostly statistically significant) from May to November with minima in June and October, while the drier part of
the year shows larger jumps between months. Instead of cloud cover, effective cloud albedo (ECA) from CM SAF satellite data
is used here. Trends in this quantity are clearly anti-correlated with the SDSR trends but the magnitude is smaller. Predomi-
nantly positive trends during FRS, LDS, and SRS are consistent with the station observations shown in Fig. 10. The difference
of the two normalized trends (golden line in Fig. 12d) points to a positive aerosol change in all months with a maximum in
September, consistent with the visibility trends at observing stations shown in Figs. 9a and 9b. Despite, the relatively weak
signal at Lamto, the overall conclusion is that there are robust indications for an aerosol-related decrease in SDSR in the entire
study region through most of the year.

## 4   Summary and conclusions

In this paper we have investigated the hypothesis that the observed recent increase in manmade aerosol pollution over SWA has
impacted seasonal rainfall trends on the decadal timescale. Given ongoing issues with climate models to realistically represent
the West African monsoon, we decided to analyze this question based on available observational records alone. Given a large
interannual variability, a strong influence from other climatic factors such as SSTs, and an overall relatively limited database,
the investigation strategy was to look at this problem from many different angles in order to reach a balance of evidence, even
if statistical significance for individual factors may be low. Trend analyses are presented both for an extended period from
1983–2015 and a shorter more recent period from 2001–2017 (with some smaller deviations due to data availability). Most
attention is given to the LDS, when the rainfall maximum is to the north of SWA, and to the SRS, while results for the FRS are
less significant and thus only provided in the Supplementary Material.

The main findings of this paper are:

– Interannual rainfall variability across most of SWA and during both LDS and SRS is positively correlated with SSTs over the tropical Atlantic, while the influence of other climatic factors is relatively weak. A multi-linear statistical model was produced to remove these effects in order to isolate residual trends that could be related to aerosol.

  – During the LDS, SWA has dried significantly with the notable exception of the coastal areas between Accra and Lagos, where urbanization may have affected the local land-sea breeze circulation. Slightly warming SSTs in the Atlantic
during the last two decades have also supported rainfall increases in the coastal areas. Removing those effects leaves a statistically significant and considerable negative trend over most of SWA that has accelerated in recent years.

  – During the SRS, SWA has become wetter in the far north and at the immediate coast with drying in between, particularly over Ivory Coast and Ghana. In the last two decades, the drying trend spread into the coastal zone and farther inland, restricting positive trends to central Ivory Coast, the northern Lake Volta area, and smaller parts of Nigeria. As in the
LDS, removing the influence of warming SSTs, creates a predominantly negative trend throughout most of SWA, but particularly in the coastal strip.

  – Satellite estimates of AOD show large increases over the tropical Atlantic to the immediate south of SWA, but over land measurements are impeded by too frequent cloud cover. These increases are related to biomass burning aerosol from Central Africa being carried into SWA, particularly during the LDS, when this import adds significantly to local sources
(Haslett et al., 2019a).

  – Station estimates of horizontal visibility suggest a substantial decrease over recent decades, particularly over coastal and low-lying inland locations. These cannot be explained by changes in cloudiness.

  – Station and satellite estimates of SDSR suggest a marked reduction throughout most of the year but particularly during the LDS and SRS. This is partly explained by an increase in cloudiness and partly through an increase in aerosol,
consistent with the visibility analysis.

  – These results cannot proof a local aerosol effect on rainfall but the balance of evidence strongly suggests such an association. Amongst the arguments to support such a claim are the geographical distribution (more pollution and rainfall suppression in the lowlands and even small improvements over higher ground), the seasonality (a larger affected area during the LDS due to faster transport, more pollution import from Central Africa and less wet deposition), and in par-
ticular the worrying acceleration of trends in the last two decades when pollution levels rose strongly (Liousse et al., 2014).

  – With respect to possible mechanisms of an aerosol impact on rainfall, this work cannot provide any new evidence but can extrapolate findings of past studies based on shorter time periods or high-resolution model results, particularly those obtained in the framework of the DACCIWA project. These suggest little influence of aerosol on cloud properties making
indirect effects rather unlikely to be the predominant pathway (Deetz et al., 2018b; Taylor et al., 2019). On the other hand, there is evidence that the high relative humidity at low levels, particularly during the LDS and SRS (Fig. 3), leads to

wet growth of aerosol particles, which strongly enhances the direct effect (Deetz et al., 2018a; Haslett et al., 2019b). Decreases in relative humidity may to some extent counteract the aerosol increase. For the LDS Kniffka et al. (2019) have already demonstrated the high sensitivity of rainfall to incoming solar radiation, supporting the idea of a direct aerosol effect. Such work does not exist for the FRS and SRS, but the rainfall classification by Maranan et al. (2018) suggests a predominance of long-lived, organized systems in the former and more locally triggered convection in the latter, possibly explaining the larger aerosol signal in SRS.

The present study has not explicitly considered changes in the land surface as a potential forcing factor of the drying trends, modulated for example through changes in evapotranspiration, albedo and storm triggering and development. For the Sahel, changes in vegetation/land cover appear to amplify decadal rainfall signals (Zeng et al., 1999; Kucharsky et al., 2013) and the vegetation feedback tends to prolong wet monsoon seasons through moisture recycling (Yu et al., 2017). Moreover, soil moisture content and gradients as well as vegetation boundaries can affect triggering and intensifying convective systems (Taylor & Lebel, 1998; Taylor et al., 2011; Hartley et al., 2016; Klein & Taylor, 2020). However, all of these effects are likely less relevant in the wetter and more densely vegetated Guinea Coastal region, for which long-term impacts are unknown.

Despite the often indirect evidence presented here and admittedly in parts qualitative argumentation, we feel that the presented results are strong and convincing enough to justify more attention to this problem. While a negative influence of a long-term increase of manmade aerosol has been claimed for other regions such as southern Africa (Hodnebrog et al., 2016) and eastern China (Huang et al., 2016), this study is the first to raise this issue for SWA. The consequences of this are twofold: First, scientists should increase efforts to better understand the mechanisms involved in the aerosol-rainfall connections using combinations of ground and satellite data in concert with models capable of representing the full complexity of the problem at hand. For the adjacent Sahel, Marvel et al. (2020) recently applied a multivariate fingerprinting technique to show that a greenhouse gas forcing signal is already detectable in the early 21st century while a forcing signal from global aerosol changes can be expected to emerge only in the middle of this century. It would be interesting to conduct a comparable study for regional aerosol in SWA. Second, policymakers in SWA are advised to prevent a further increase in air pollution through suitable regulations and improved technology (Evans et al., 2018). This paper has shown that the aerosol-induced rainfall suppression is significant (order several 10s of mm per month) and has been accelerating in the last two decades, although some of the effect has been concealed by opposing effects from SST changes. Allowing the air pollution problem to further deteriorate in the future could therefore cause significant socio-economic damage through impacts on human health and water supply, which in turn is closely linked to food security and energy production in SWA.

*Data availability.* ERA5 data (Copernicus Climate Change Service, 2017; Hersbach et al., 2020) are available via https://cds.climate. copernicus.eu. CHIRPS data (Funk et al., 2015a) can be downloaded from ftp.chg.ucsb.edu/pub/org/chg/products/CHIRPS-2.0/global_daily/ netcdf/p25/. KASS-D rainfall data and observations from Lamto can be requested from the third author. Data ownership restrictions apply. MIDAS surface observations (Met Office, 2006) are available at http://catalogue.ceda.ac.uk/uuid/0ec59f09b3158829a059fe70b17de951. Radiation time series from Parakou are available via the baobab web page http://baobab.sedoo.fr/DACCIWA/?editDatsId=1785, Fink A. and

585 M. Marlon (2019) "CNR1_ Cotonou_ Parakou." SEDOO OMP. doi: 10.6096/baobab-dacciwa.1785. CM SAF satellite data (Pfeifroth et al., 2017) can be downloaded from https://doi.org/10.5676/EUM_SAF_CM/SARAH/V002. MODIS AOD data (Platnick et al., 2017) are available at http://dx.doi.org/10.5067/MODIS/MOD08_M3.061 and http://dx.doi.org/10.5067/MODIS/MYD08_M3.061. Climate indices from https://psl.noaa.gov/gcos_wgsp/Timeseries/Data/nino34.long.data for Niño3.4, from https://psl.noaa.gov/data/timeseries/monthly/AMM/ for AMM and from http://www.esrl.noaa.gov/psd/data/timeseries/AMO/ for AMO are used in this study while the Atlantic 3 and Indian Ocean

indices are computed using HadISST data which were obtained from https://www.metoffice.gov.uk/hadobs/hadisst/ and are © British Crown Copyright, Met Office, 2020, provided under a Non-Commercial Government Licence http://www.nationalarchives.gov.uk/doc/non-commercial-governme version/2/.

*Author contributions.* G. P. did most of the data analysis and plotting while P. K. initially formulated the idea for this study. A. H. F. contributed with detailed knowledge about available data sets and their interpretation and A. K. analyzed station-based radiation data. All

595 authors jointly discussed the results and wrote the paper.

*Competing interests.* The authors declare no competing interests.

*Acknowledgements.* The DACCIWA project has received funding from the European Union Seventh Framework Programme (FP7/2007-2013) under grant agreement no. 603502 (EU project DACCIWA: Dynamics–aerosol–chemistry–cloud interactions in West Africa). Radiation measurements at Parakou were carried out by the IMPETUS project funded by the BMBF project IMPETUS (BMBF grant 01LW06001A,

North Rhine-Westphalia grant 313-21200200). We wish to thank Adama Diawara, Fidèle Yoroba and Veronique Yoboué for providing the radiation and cloudiness observation data from the Lamto Geophysical Observatory in Ivory Coast. The authors also thank various colleagues and weather services that have over the years contributed to the enrichment of the KASS-D database. We like to acknowledge the Analyse Multidisciplinaire de la Mousson Africaine-Couplage de l'Atmosphère Tropicale et du Cycle Hydrologique (AMMA-CATCH), which provided radiation data for Djougou. We thank one anonymous reviewer and the editor Ademe Mekonnen for their clear and thoughtful

comments, which helped us to improve our manuscript.

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

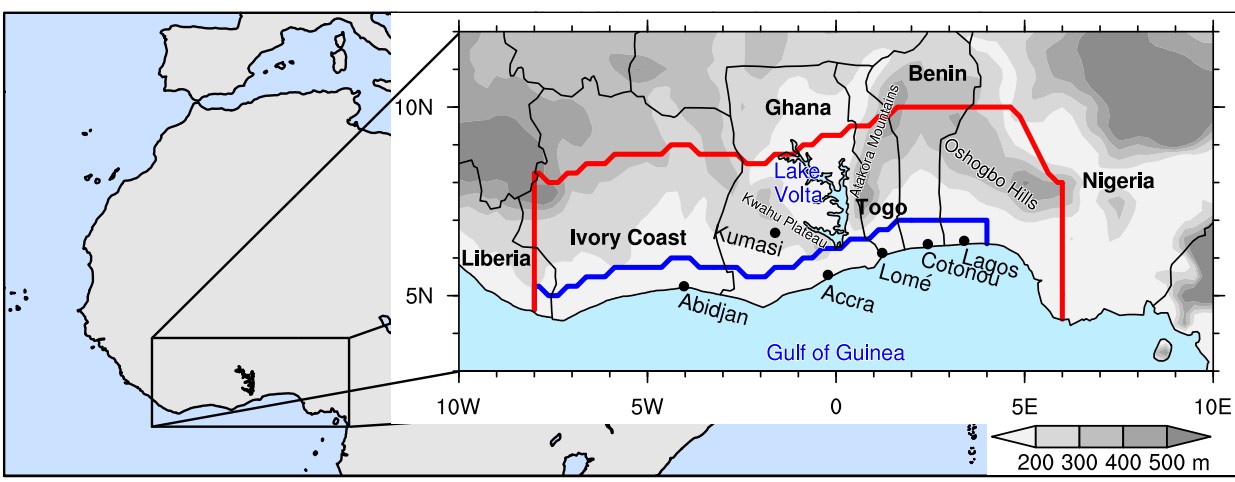

**Figure 1. Geographical overview.** Map of southern West Africa showing major cities, topographic features and the areas used for spatial averaging in this paper. The red bordered region is analyzed for the little dry season (LDS) while only the blue bordered region, referred to as "coastal strip", is considered for the second rainy season (SRS).

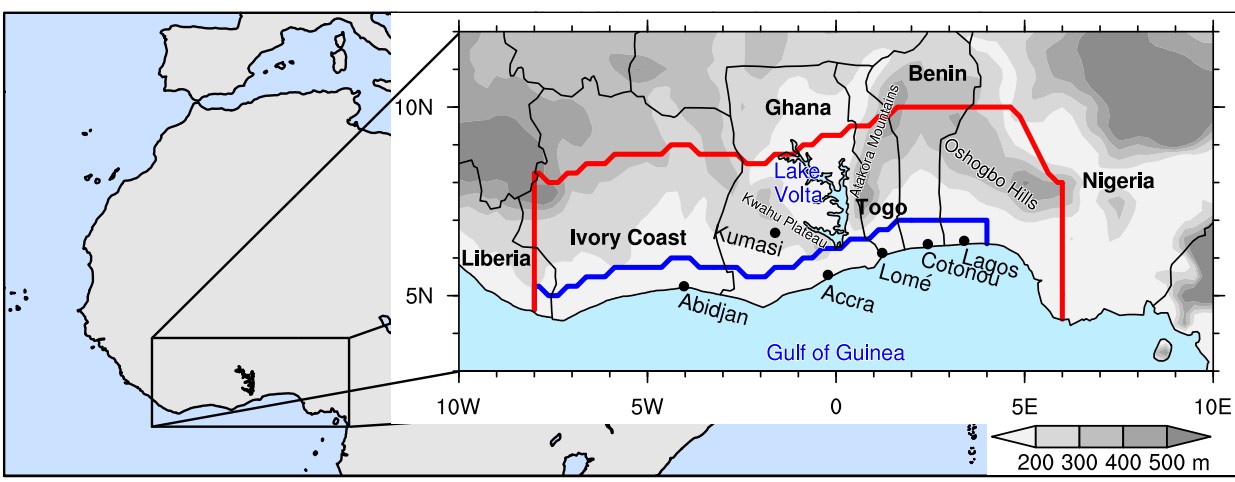

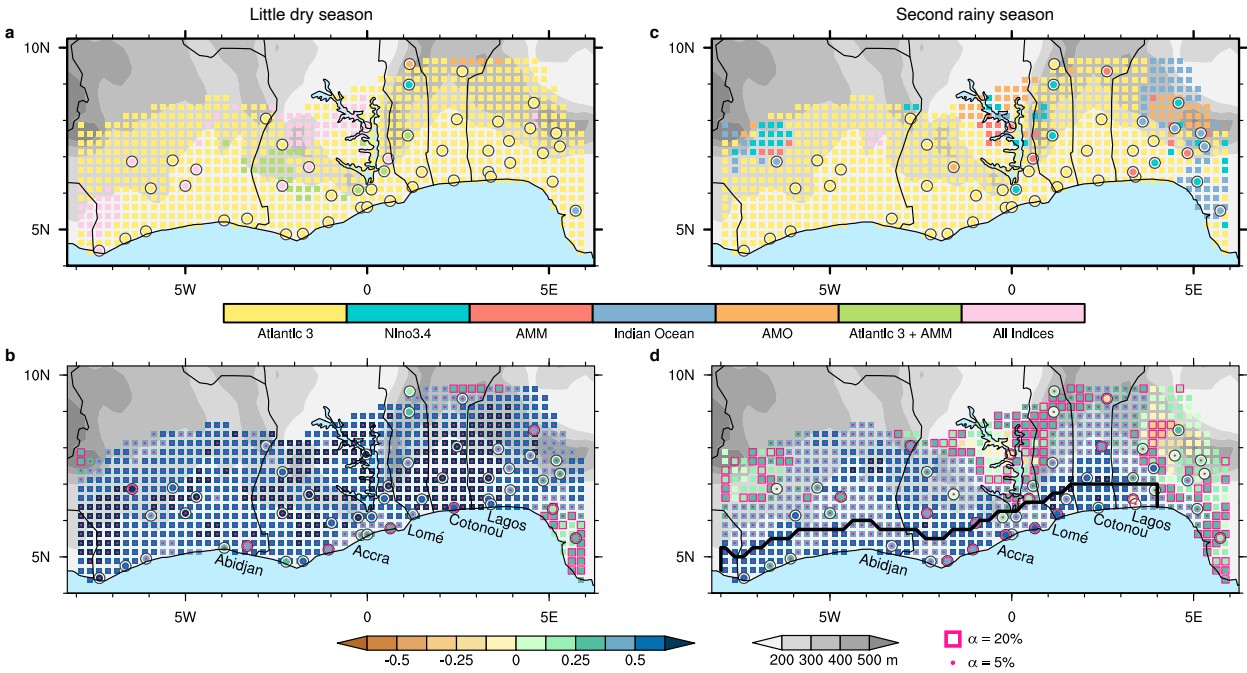

**Figure 2. Optimal linear model. a**, Combination of climate indices yielding the best results with respect to the leave-one-out cross validation procedure as described in section 2.2 for the LDS. Each of the single-index models (Atlantic 3, Niño3.4, AMM, AMO) can be found. The only combinations of indices with better results are Atlantic 3 with AMM and the combination of all five indices. **b**, Pearson's correlation coefficient of the optimal linear model with the original rainfall time series for the LDS. **c**, **d**, as a, b but for the SRS. The coastal strip is marked by a black line in d. All panels show data from 1983–2015 for the CHIRPS (colored pixels) and station data (colored circles). Statistically significant correlations in b and d are marked with pink borders of pixels and stations for the $\alpha = 20\%$ level and a pink dot in the middle for the $\alpha = 5\%$ level, respectively.

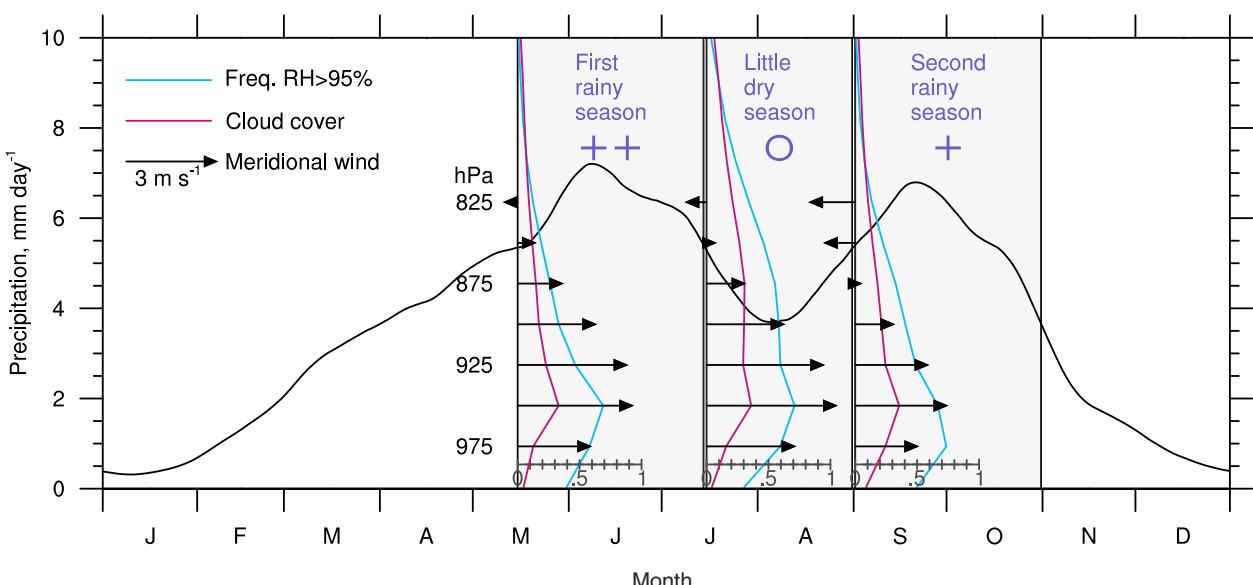

**Figure 3. Climatological overview.** Average annual cycle of meteorological conditions over southern West Africa (red bordered region in Fig. 1). Black solid line shows mean rainfall (CHIRPS), while the profiles give average conditions (ERA5, mean 06 UTC values from 1983–2015) for meridional wind (southerly (northerly) wind as vectors to the right (left), strength according to the reference vector), cloud cover (pink) and the frequency of relative humidity exceeding 95% (cyan) for the first rainy season (FRS, 15 May–14 July), the little dry season (LDS, 15 July–31 August) and the second rainy season (SRS, 1 September–31 October). Cloud cover and relative humidity range from 0 to 1 according to the scales at the bottom for each season. Symbols below the seasons' names mark the degree of convective organization (++ strong, + moderate, ○ weak) according to Maranan et al. (2018).

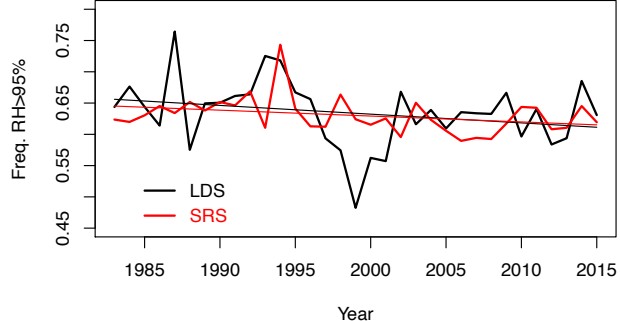

**Figure 4. Time series of relative humidity.** Time series from 1983–2015 and simple linear regression lines show the temporal evolution of the frequency of relative humitidy exceeding 95% averaged from 925–975 hPa. Data are shown for the LDS (black) and the SRS (red) and are spatially averaged over the entire study region (see Fig. 1).

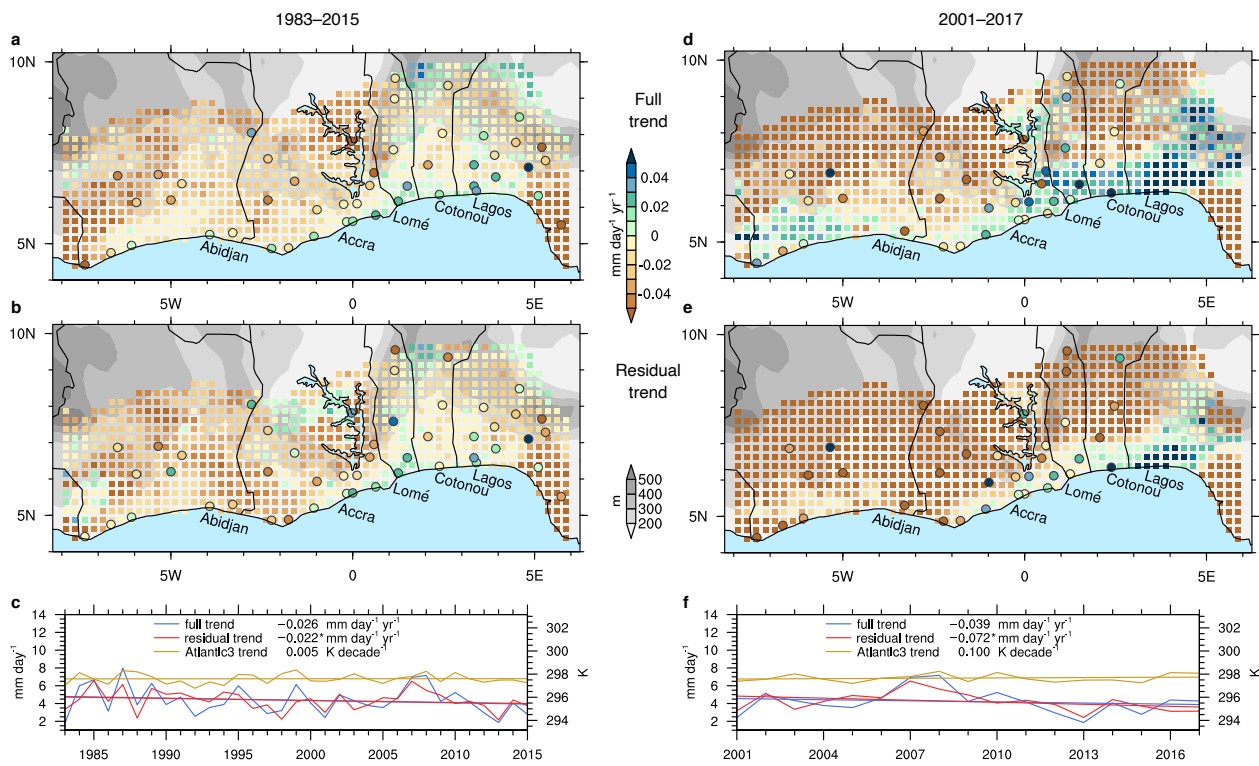

**Figure 5. Rainfall trends during the little dry season. a**, **b**, **c**, trends in rainfall from 1983–2015 for CHIRPS data (colored pixels) and station data (colored circles). **d**, **e**, **f**, as a, b, c but for recent trends from 2001–2017. Shown are full (a, d) and residual (b, e) trends. Note the overall good agreement between CHIRPS and station data. Topography is shown in grey shadings. "Residual" in this context means that a statistical model was used to remove the influence of other known climatic factors on rainfall (for details, see section 2.4). Time series (c, f) of full (blue) and residual (red) CHIRPS rainfall, spatially averaged over the entire region and the Atlantic 3 climate index (gold) are shown together with their trends over the respective time span. A * in c and f marks statistically significant trends on the $\alpha = 20\%$ level.

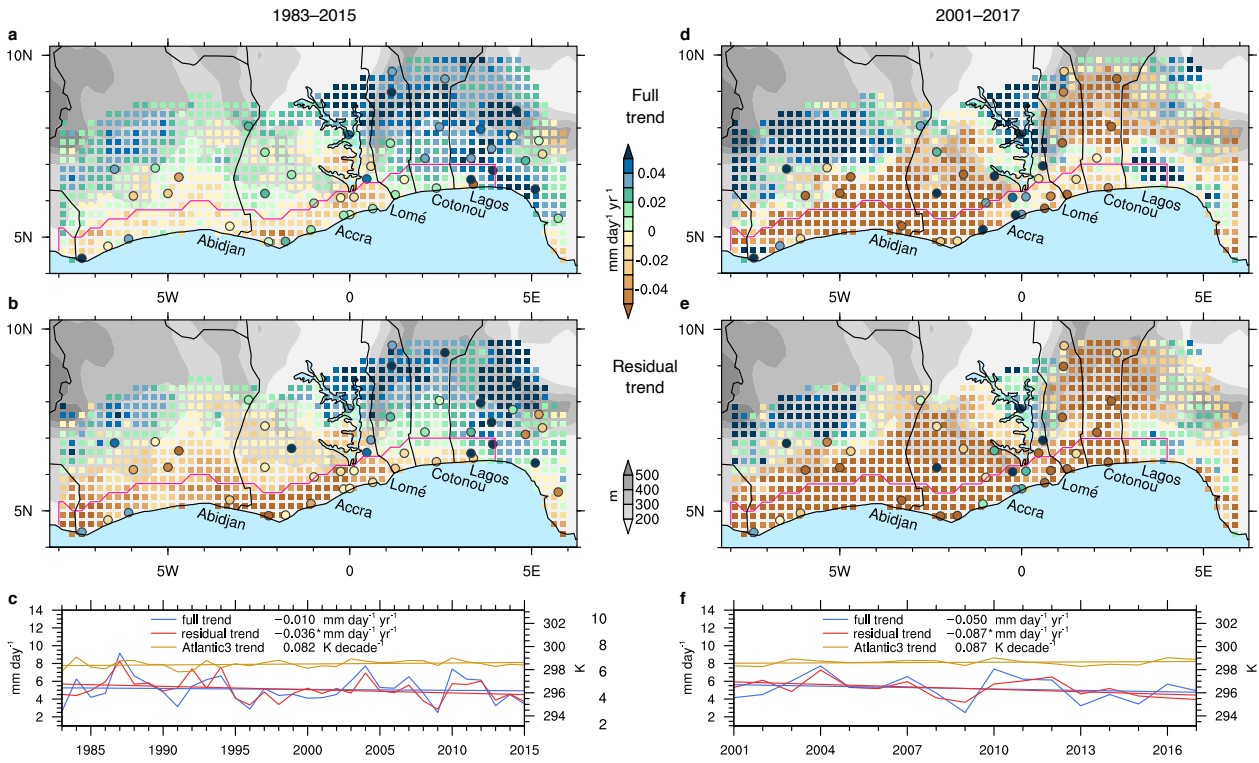

**Figure 6. Rainfall trends during the second rainy season. a**, **b**, **c**, trends in rainfall from 1983–2015 for CHIRPS data (colored pixels) and station data (colored circles). **d**, **e**, **f**, as a, b, c but for recent trends from 2001–2017. Shown are full (a, d) and residual (b, e) trends. Note the overall good agreement between CHIRPS and station data. Topography is shown in grey shadings and the coastal strip is marked by a pink line. "Residual" in this context means that a statistical model was used to remove the influence of other known climatic factors on rainfall (for details, see section 2.4). Time series (c, f) of full (blue) and residual (red) CHIRPS rainfall, spatially averaged over the entire region and the Atlantic 3 climate index (gold) are shown together with their trends over the respective time span. A * in c and f marks statistically significant trends on the $\alpha = 20\%$ level.

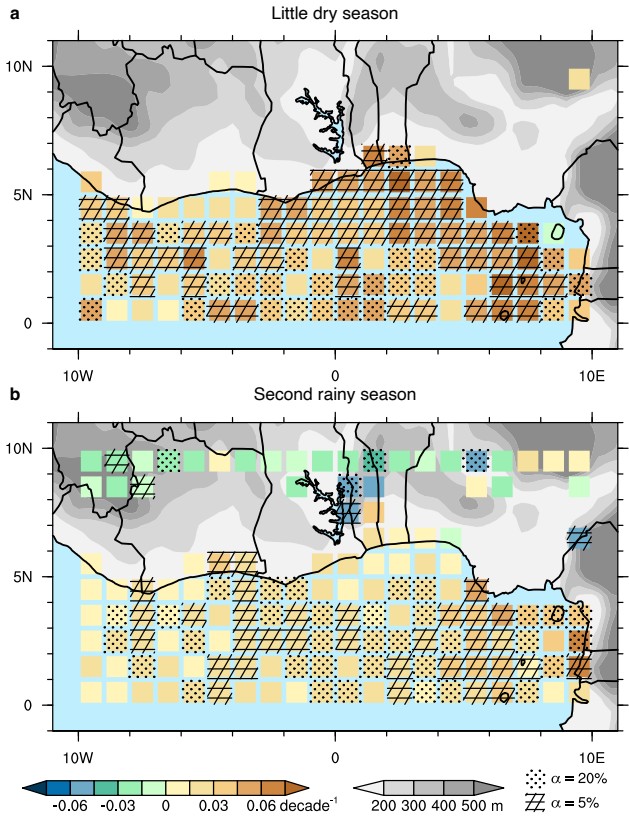

**Figure 7. Aerosol trends. a**, **b**, Trends in MODIS aerosol optical depth (AOD) from 2002–2018 calculated from the mean values of Aqua and Terra monthly AOD for the LDS (a) and the SRS (b). Only pixels with a sufficiently complete record are displayed here (see section 2.3 for further details). Statistically significant trends are marked with dotted ($\alpha = 20\%$) and hatched ($\alpha = 5\%$) pixels, respectively.

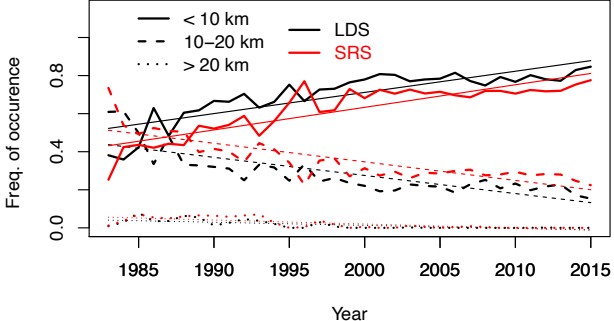

**Figure 8. Time series of horizontal visibility.** Time series from 1983–2015 and simple linear regression lines show the temporal evolution of the frequency of horizontal visibility observations for the three categories "below 10 km" (solid), "between 10 and 20 km" (dashed), and "above 20 km" (dotted). Data are shown for the LDS (black) and the SRS (red) and are spatially averaged over all available stations (see Fig. 9).

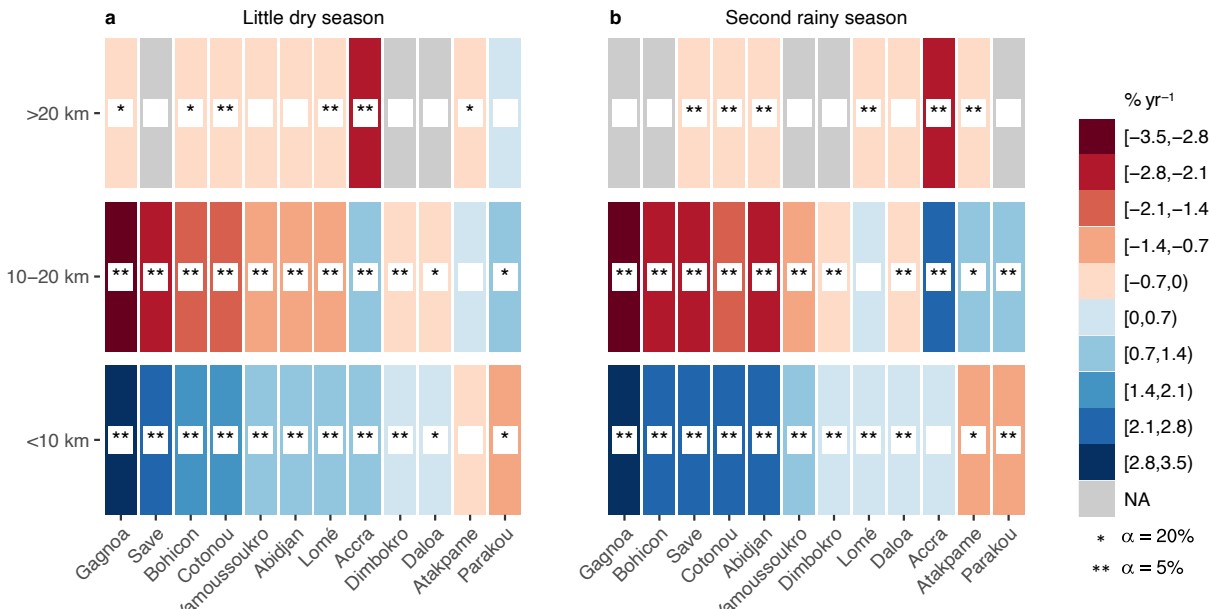

**Figure 9. Trend in horizontal visibility.** Trend in horizontal visibility based on station data from 1983–2015 for the LDS (**a**) and the SRS (**b**). Shown are the trends of the frequency of horizontal visibility observations for the three categories "below 10 km" (bottom), "10–20 km" (middle), and "above 20 km" (top) in % yr$^{-1}$. Stations are ordered from left to right according to the strength of the increase in the "below 10 km"-category. In both panels a * (**) marks statistically significant trends on the $\alpha = 20\%$ (5%) level.

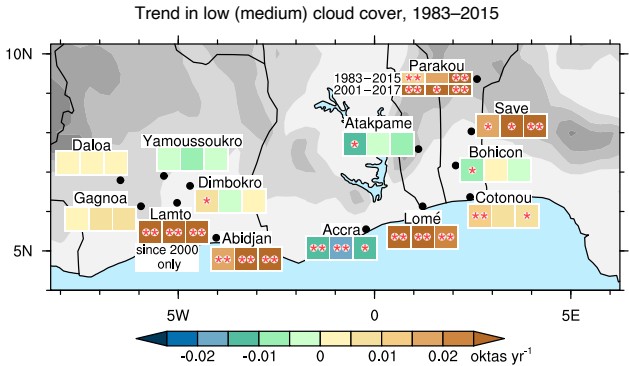

**Figure 10. Trend in cloud cover.** Trend in low and medium cloud cover based on station data from 1983–2015. Three squares for each station show trends for the three seasons FRS (left), LDS (centre) and SRS (right). For Parakou the second row shows the trend from 2001–2017 as in Fig. 12. For Lamto trends in total cloud area fraction between 2000 and 2015 are shown. Topography is shown in grey shadings. A * (**) marks statistically significant trends on the $\alpha = 20\%$ (5%) level.

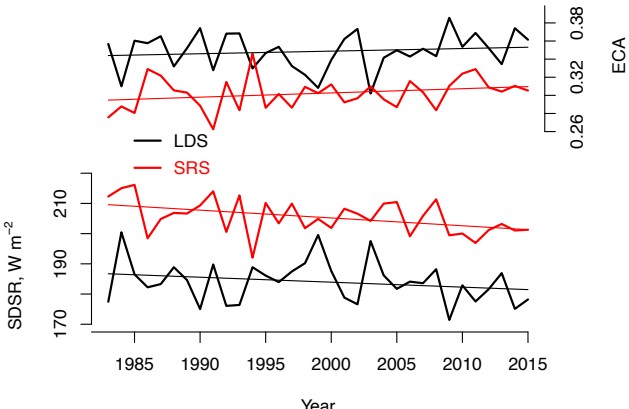

**Figure 11. Time series of cloud cover and radiation.** Time series from 1983–2015 and simple linear regression lines show the temporal evolution of the CM SAF satellite data of effective cloud albedo (ECA) and surface downwelling shortwave radiation (SDSR). Data are shown for the LDS (black) and the SRS (red) and are spatially averaged over the entire study region (see Fig. 1).

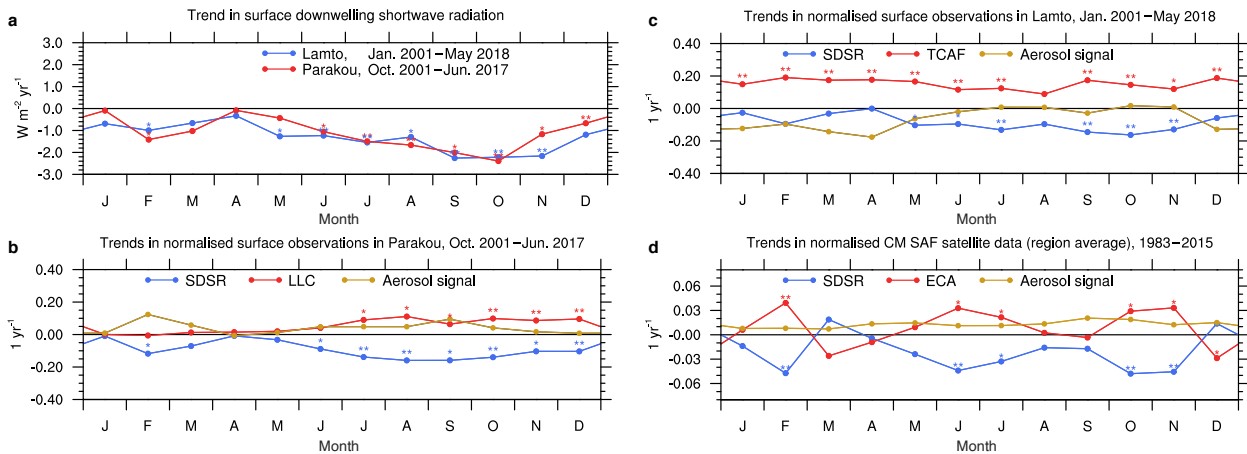

**Figure 12. Trends in radiation. a**, Annual cycle of trends in surface downwelling shortwave radiation (SDSR) at Lamto and Parakou from 2001–2018 (see legends in panels for exact dates). **b**, **c**, Normalized trends of SDSR and cloud cover at Parakou (b) and Lamto (c). An aerosol signal (gold) is derived from SDSR and cloud cover (see section 2.5 for details). While both stations measure SDSR, low and medium cloud cover (LLC) is only observed at Parakou. For Lamto the total cloud area fraction (TCAF, see section 2.4) is used to calculate the aerosol signal. **d**, As b, c but for CM SAF satellite data averaged over the entire study region (see Fig. 1). Instead of cloud cover the effective cloud albedo (ECA) is used to calculate the aerosol signal. In all panels a * (**) marks statistically significant trends on the $\alpha = 20\%$ (5%) level.