# Peer review of "Increasing manmade air pollution likely to reduce rainfall in southern West Africa"

_Atmospheric Chemistry and Physics, 2020_

## Referee Comment (RC1) · Anonymous Referee #1 · 13 Jul 2020

General comments

The paper focuses on rainfall trends in southern West Africa from the 80s to present day. After removing the effect of regional and global SST forcing, the authors analyse the effect of aerosol on the "residual trend" of precipitation. By using direct observation and indirect assessment of atmospheric aerosol content, the authors show that the "residual" drying trend in the region can be explained by the increasing trend in air pollution.

The paper is very well written, existing literature is properly cited, motivation and objectives clearly stated, methods are correctly applied and explained in detail. Results follow from evidence, and the limitations of the study are discussed.

[Figure]

As a reviewer, I first want to thank the authors for providing such a high level manuscript, which I really enjoyed to read for its clarity and rigor in the development of discussion. Unfortunately it is not always the case.

The paper adds an important piece of knowledge in a crucial topic such as the environmental effects of increasing air pollution in southern West Africa, which is still underrepresented in the atmospheric science literature. Although evidence for this effect is only indirect, the paper has the merit to highlight a possible/likely link and invite to further investigate. Finally, I definitely recommend the paper for publication.

Specific comments

However, I invite the authors to undertake few changes in the way they present the conclusions of the study, to make it even more rigorous. Although evidence is shown that the drying trend is accompanied by an increase in atmospheric aerosols in the region, no physical evidence is presented in the paper that this is at the origin of the drying trend. Moreover, the effect of manmade pollution is not quantified: no local sources of pollution are analysed, and a quantification of the ratio of manmade vs natural biomass burning aerosols is not presented. I recognise that all these aspects are openly and clearly discussed in the main text, however I feel that the title and abstract of the paper could be misleading in this sense. I'd suggest first to change the title in something sounding like: "Indications for increasing air pollution likely to reduce rainfall in southern West Africa". I also suggest to make clear in the abstract that a quantification of the manmade impact is not possible at this stage.

In addition, I also believe that other possible forcings for the drying trend should be mentioned. For instance, the role of vegetation and land cover in controlling precipitation in the Sahel is well known (see e.g. https://link.springer.com/article/10.1007/s00382-012-1397-x and https://www.nature.com/articles/s41467-017-02021-1), and a possible role for southern West Africa should be discussed.

Few minor points are listed below.

L90: "The two rainy seasons were not studied by DACCIWA, but compared to the LDS the climatology of wind and rainfall suggests a lesser import of aerosol from Central Africa, less spreading of coastal pollution inland with the monsoon winds, and more wet deposition due to the enhanced rainfall". Any reference?

L93: I feel that "equally" suggests that the increase in biomass burning aerosol equals the increase in anthropogenic emissions. Please rephrase.

L175: please add a short description of the Sen's slope method.

L305: I'd say that "indication" can be get from a personal communication, rather than "evidence".

L428: not clear to me the reference to the political situation (too vague) in this sentence, please clarify.

Trends significance: in the main text and captions, please explicitly mention when trends are not significant.

Figure 2: any indication on the statistical significance of the correlation coefficients?

Figure 5: note that positive trends inland could be associated with the observed delay in the retreat of the Sahelian rainy season in the last decades (see e.g. https://rmets.onlinelibrary.wiley.com/doi/full/10.1002/joc.4638). This could also be associated with the reduction in aerosols (see L410).

Technical corrections

L339: missing reference.

Figure 7: what do red stars mean? (significance I guess)

[Figure]

---

## Editor Comment (EC1) · Ademe Mekonnen (Editor) · 15 Aug 2020

Comments "Increasing manmade air pollution likely to reduce rainfall in southern West Africa", by Pante et al.

I commend the authors for taking up an important problem that is understudied.

I recommend minor revision, but I expect authors to address concerns below.

I would suggest up front that the paper should go through critical review to improve grammar and direct "usage" at places. For example, ". . . not inconsistent . . . (Line 424)" can be re-written using a direct expression ". . . consistent . . .". Indirect explanation may lead to misunderstanding. Also, I suggest authors should be careful on some "throw away" phrases such as "traditionally ... (Line 47) and " ... hardly organized convection

[Figure]

. . . (Line 44)". I encourage authors to re-write sentences framed in such way at few places.

Abstract The abstract, in this reviewer's opinion, should incorporate the overarching objective of the work, most important data and methods, and major conclusions. The first few sentences of the abstract are vague and would require either reference or additional explanation. These broad summaries may be removed. Below are examples.

L1-2: "Southern West Africa has one of the fastest growing populations worldwide. This has led to a higher water demand and lower air quality." No evidence and no clear reference. Fast population growth→ more energy demand→ more pollution can be discussed in the body of the paper. This reviewer did not find investigation on population increase and implication of this in the body of the paper.

L3: "little" dry season. What does that mean? Is there "large" dry season? This is mainly a terminology issue. I encourage the authors to consider changing to an appropriate phrase. I am not sure if "short" could substitute "little". Terms "Short rain season, longer rain season, etc." are widely used in African monsoon literature.

L5-7: "Increased pollution"→ "dimming solar radiation"→ "suppressing rainfall" – For a "linear" reader this would be hard to understand. I would encourage more clarity.

L10-12: " . . . decreases in horizontal visibility and incoming surface solar radiation are consistent with the hypothesized pollution impact . . ." . I would defer this to later sections. Or, authors may consider adding clarity.

Sections 1-3: Line 19-25: I encourage authors to consider to significantly reduce or improve the introduction. Most of the statements are not relevant. Just a couple of sentences to indicate the relationship between increasing population and energy demand and what that means to vulnerability. The overarching objective of the paper is explain rainfall trend and variability and the role of pollution resulting from this.

L27: "Generally, the climate of SWA is strongly controlled by the seasonal evolution

of the West African monsoon. Given the proximity to the equator, ....". This is vague. SWA is part of W. African monsoon (note: "Generally" is used). There are several examples that can show bi-modal, tri-modal rainfall away from the equator (about 10o away from Equator). As the authors know very well, the reason low-latitude regions exhibit bimodal or multimodal rainfall pattern are associated with north-south swing of the convective zone (ITCZ in the earlier literature). For example, L28 – dry season Dec-Feb. is related to the convective zone far to the south. I think the month-to-month variability should be described in that context and as described in along lines 31-34.

L44: I recommend extra care in the use of adjectives or modifiers. What does "hardly organized convection" mean? Weakly organized? Also see comments above. Considering proximity of SWA to sea, and given the description in lines 35-45, what is the role of land-sea breeze?

Overall: the first few paragraphs of the Introduction may refocus on trend and mechanisms that potentially explain the trend and setting the stage for "pollution effect".

L47: Again, I suggest remove "traditionally" and rephrase the sentence. Something like . . . " In the literature, marked SWA rainfall variability on interannual to decadal timescale is linked to . . .."

L50" "...undergone a mild recovery of rainfall since then . . ..". Can you be more specific? In the 1990? 2000s?

L54-56: "While SST changes appear to have played a role in creating this trend, the seasonality and magnitude of changes remains poorly understood ..." Any reference? This inference is drawn after discussion of "interannual and decadal" variability and trend. It is unclear how is "seasonality" can be discussed in the same line(s).

In the data and methods section, various techniques of trend analysis are discussed. Various parameters that have influence on rainfall have been explored. One way of investigating a quick and simple method to identify most important impact on a predictand is to run a partial correlation analysis. I'm not suggesting to redo the results, but I would like to know if this was considered. The reason I suggest this is that often times two predictors can have strong relationship and can affect one another and analysis based on linear relationship can give biased results.

Association between AOD and rainfall: I would expect a timeseries plot that shows how radiation, cloud cover and horizontal visibility including relative humidity (RH) > 95% vary over time. Before discussing trends, a reader would have a chance to form an idea about how these parameters change over time. I believe this could be justified. This could be shown in section 3.1.

Lines 275: Discussion can be made clearer if you refer to Fig. 3.

Yes, The RH >95% is in LDS deeper compared with other seasons. Deeper RH may be associated with stronger meridional winds (winds are also stronger at lower levels of the troposphere). So, more moist air is brought into (or circulating) the region from ocean. But, this may not be translating into rainfall. Would a local dynamic factor lacking? As shown, LDS is characterized by weak convective organization. Climatologically the convective zone is far to the north. Please explain.

Line 280: Is 06UTC closer to cloud maximum? Since the solar time for the study area is almost the same as the Greenwich time, 06UTC is 6AM local solar time. If my assumption is true, cloudiness over SWA is more like the oceanic condition. Does information for cloud cover is more from more stratified clouds? I thought convective cloud coverage is later in the day for coastal areas (solar noon).

Lines 335-340: Question mark after a reference. Indicates something is missing.

In the Introduction section, there is a discussion related to aerosol transport from out of the study area. Although qualitative descriptions are provided, Aerosol transported from out of the region is not quantified or shown. In lines along line 340-350, since the trend can not be fully explained by the parameters, the conclusion was "anthropogenic aerosols play a role". This is unclear. How do we know whether anthropogenic aerosols play the main role in the rainfall trend? There could be multitude of factors that could explain the patterns. I guess the problem is the relationship between variables identified and aerosols not firmly established ("indirectly" to borrow authors words). I recommend adding a sentence or two explanation to help readers. Explain how "inland transport" might occur (in absence of data, literature might help).

Lines 378-382: This paragraph is interesting. The authors stated that MODIS AOD data is either "not useful" or very limited. Given this justification, I'm wondering why authors decide to continue using it? I suggest the rational for continuing the investigation is unclear. I encourage authors to re-write this.

Section 3.3.2: Visibility and cloud cover As described in the text, the source of visibility measurements are observers. That means this data is prone to biases. One person's visibility estimation could be different form another person's estimate (different people taking observations over time). I am curious if authors consider a different way of looking at the same data. That is, (1) categorize visibility estimates into ranges, e.g., low, medium and high, then (2) workout the frequencies of each category. For example, what is the frequency of low visibility? Then, this can be used to describe the objective of the research. Higher frequency of low visibility can be related to high frequency of fog or bad weather, or pollution. Trend analysis can be performed on frequencies. The advantage of this approach is reducing biases because high and low visibility are categories and do not directly point to a measurement. Interpretation would also be easier. I thought about this because of the following!

I have hard time understanding a 200m/yr reduction in visibility (Line 420). What is the difference between an average 1km low visibility and average 0.8km visibility? How do this be a proxy for aerosols?

Figures: Captions are well explained but can be shortened at places. For example, in Fig. 3 2.0 and ERA5 can be used while omitting their full description. Meridional

wind depiction is confusing. By plotting it as a vector, you are assuming zero zonal wind. Since you are considering one component of the wind field, you may consider using a line plot (with solid for positive and dashed for negative). Describe or give context to symbols just below season captions (e.g., ++ higher convection, O less, and + moderate)

---

## Author Comment (AC1) · 25 Sep 2020

First review of the manuscript

"Increasing manmade air pollution likely to reduce rainfall in southern West Africa"

submitted to Atmos. Chem. Phys. Discuss., https://doi.org/10.5194/acp-2020-463-RC1

by Gregor Pante et al.

*We thank the reviewer and editor for their clear and thoughtful comments, which helped us to improve our manuscript. Please find our replies in red italics below.*

**Anonymous Referee #1**

General comments

The paper focuses on rainfall trends in southern West Africa from the 80s to present day. After removing the effect of regional and global SST forcing, the authors analyse the effect of aerosol on the "residual trend" of precipitation. By using direct observation and indirect assessment of atmospheric aerosol content, the authors show that the "residual" drying trend in the region can be explained by the increasing trend in air pollution.

The paper is very well written, existing literature is properly cited, motivation and objectives clearly stated, methods are correctly applied and explained in detail. Results follow from evidence, and the limitations of the study are discussed.

As a reviewer, I first want to thank the authors for providing such a high level manuscript, which I really enjoyed to read for its clarity and rigor in the development of discussion. Unfortunately it is not always the case.

The paper adds an important piece of knowledge in a crucial topic such as the environmental effects of increasing air pollution in southern West Africa, which is still underrepresented in the atmospheric science literature. Although evidence for this effect is only indirect, the paper has the merit to highlight a possible/likely link and invite to further investigate. Finally, I definitely recommend the paper for publication.

Specific comments

However, I invite the authors to undertake few changes in the way they present the conclusions of the study, to make it even more rigorous. Although evidence is shown that the drying trend is accompanied by an increase in atmospheric aerosols in the region, no physical evidence is presented in the paper that this is at the origin of the drying trend. Moreover, the effect of manmade pollution is not quantified: no local sources of pollution are analysed, and a quantification of the ratio of manmade vs natural biomass burning aerosols is not presented. I recognise that all these aspects are openly and clearly discussed in the main text, however I feel that the title and abstract of the paper could be misleading in this sense. I'd suggest first to change the title in something sounding like: "Indications for increasing air pollution likely to reduce rainfall in southern West Africa". I also suggest to make clear in the abstract that a quantification of the manmade impact is not possible at this stage.

*We perfectly understand and share your desire for more physical evidence but we think that this could only be provided through modelling experiments. Given the issues models have with the complex West African meteorology and atmospheric composition, we think that there is room and justification for a purely observational study. This point was already explained in the original manuscript. We argue that we do what can be done with the existing data on the considered time-space scale – and given the issues in data availability and quality that we discuss at length in the paper.*

*On the question of local sources of pollution: The existing long-term data basis is very thin and we therefore think that there is little point in trying to add on to established studies such as Liousse et al. (2014).*

*On the question of manmade vs. natural biomass burning aerosols: In the moist tropics there are relatively little natural fires, so most of what we see today is manmade. We now articulate this in one sentence in the introduction in the following way: "which is affected by the advection of burning aerosol from mostly manmade agricultural and forest fires in Central Africa". It would be interesting to distinguish between local West African and imported Central African aerosol but again there is no data source to quantify this. As we show in the paper, the MODIS AOD gives at least some good indications of how the import has changed over the years. For the rest, we have to rely on Liousse et al.*

*With respect to the title: We think that the „likely" already expresses the uncertainty you mention and prefer not to change it.*

*With respect to the abstract: We have reworded parts to more clearly express the indirect evidence and the lack of quantification as follows: "Over southern West Africa, no long-term aerosol records are available inhibiting a direct quantification of the manmade effect."*

In addition, I also believe that other possible forcings for the drying trend should be mentioned. For instance, the role of vegetation and land cover in controlling precipitation in the Sahel is well known (see e.g. https://link.springer.com/article/10.1007/s00382-012-1397-x and https://www.nature.com/articles/s41467-017-02021-1), and a possible role for southern West Africa should be discussed.

*We agree that land cover signals can be significant in semi-arid regions like the Sahel. However, in the moister southern part of West Africa they will likely be less relevant. For the Sahel, it is well known that vegetation feedbacks can amplify decadal rainfall imposed by changes of sea surface temperatures (SSTs) in tropical ocean basins and the Mediterranean Sea (e.g., Zeng et al. 1999; Kucharsky et al 2013). We also appreciate the reference to Yu et al. (2017) who show in a statistical approach using various observational data sets that the vegetation feedback may dominate the SST forcing in the late/post-monsoon season, i.e. in September–November, thus largely covering the second rainy season at the Guinea Coast. Yu et al. (2017) suggest that the vegetation effect is related to moisture recycling augmenting the frequency of rainfall events, whereas Kucharsky et al. (2013) physically relate the positive vegetation feedback to the albedo effect. Both effects are likely less relevant in the more densely vegetated coastal zone. Hartley et al. (2016) suggest effects of tree-grass boundaries on the initiation and intensity of convective systems in West Africa, yet no study on the impact of vegetation heterogeneities on rainfall trends are known to the authors for southern West Africa.*

*Nevertheless, we have now mentioned this point more explicitly in the discussion: "The present study has not explicitly considered changes in the land surface as a potential forcing factor of the drying trends, modulated for example through changes in evapotranspiration, albedo and storm triggering and development. For the Sahel, changes in vegetation/land cover appear to amplify decadal rainfall signals (Zeng et al. 1999; Kucharsky et al. 2013) and the vegetation feedback tends to prolong wet monsoon seasons through moisture recycling (Yu et al. 2017). Moreover, soil moisture content and gradients as well as vegetation boundaries can affect triggering and intensifying convective systems (Taylor and Lebel 1998; Taylor et al. 2011; Hartley et al. 2016; Klein et al. 2020). However, all of these effects are likely less relevant in the wetter and more densely vegetated Guinea Coastal region, for which long-term impacts are unknown."*

Few minor points are listed below.

L90: "The two rainy seasons were not studied by DACCIWA, but compared to the LDS the climatology of wind and rainfall suggests a lesser import of aerosol from Central Africa, less spreading of coastal pollution inland with the monsoon winds, and more wet deposition due to the enhanced rainfall". Any reference?

*We added a reference to Giglio 2006 (seasonality of biomass burning) and to Fig. 3. These statements are based on the climatological seasonal cycles of wind and rainfall shown there.*

L93: I feel that "equally" suggests that the increase in biomass burning aerosol equals the increase in anthropogenic emissions. Please rephrase.

*We replaced equally with additionally.*

L175: please add a short description of the Sen's slope method.

*The following short description was added in the revised version: "It calculates slopes for each pair of two consecutive years in a time series and the final trend is the median from all slopes."*

L305: I'd say that "indication" can be get from a personal communication, rather than "evidence".

*Changed as suggested.*

L428: not clear to me the reference to the political situation (too vague) in this sentence, please clarify.

*Rephrased: "The former appears rather unlikely, as political efforts to reduce local emissions are still in its infancy across the region and also given the overall population increase."*

Trends significance: in the main text and captions, please explicitly mention when trends are not significant.

*In addition to several places where this was mentioned already, e.g.*

- *"Averaging CHIRPS trends across the coastal strip leads to an insignificantly small trend in the full time series…" L359*
- *"These are marginally statistically significant…" L416*

*we now describe insignificant trends in more places, e.g.*

- *"…not significantly for "above 20 km…" L445*
- *"The two LDS time series reveal strong interannual variations and therefore trends are not significant on the tested α-levels (Fig. 11)." L470*

*Regarding the figure captions, we think that it is not necessary to explicitly mention that values without a \* symbol are not significant at the tested levels.*

Figure 2: any indication on the statistical significance of the correlation coefficients?

*In the revised version of Fig. 2 we added markers for the statistical significance of the correlation coefficients.*

Figure 5: note that positive trends inland could be associated with the observed delay in the retreat of the Sahelian rainy season in the last decades (see e.g. https://rmets.onlinelibrary.wiley.com/doi/full/10.1002/joc.4638). This could also be associated with the reduction in aerosols (see L410).

*Thanks for this reference. We mention this now in the discussion of Fig. 6: "Positive trends in the northern parts of the region during the SRS are consistent with a projected increase of rainfall in the Sahel associated with a delayed withdrawal of the WAM (Monerie et al., 2016)."*

Technical corrections

L339: missing reference.

*Corrected.*

Figure 7: what do red stars mean? (significance I guess)

*Indeed! We added this to the figure caption.*

**Editor comment by Ademe Mekonnen**

Comments "Increasing manmade air pollution likely to reduce rainfall in southern West Africa", by Pante et al.

I commend the authors for taking up an important problem that is understudied.

I recommend minor revision, but I expect authors to address concerns below.

I would suggest up front that the paper should go through critical review to improve grammar and direct "usage" at places. For example, ". . . not inconsistent . . . (Line 424)" can be re-written using a direct expression ". . . consistent . . .". Indirect explanation may lead to misunderstanding. Also, I suggest authors should be careful on some "throw away" phrases such as "traditionally ... (Line 47) and " ... hardly organized convection. . . (Line 44)". I encourage authors to re-write sentences framed in such way at few places.

*We changed the mentioned expressions and some more of such phrases in other places.*

Abstract The abstract, in this reviewer's opinion, should incorporate the overarching objective of the work, most important data and methods, and major conclusions. The first few sentences of the abstract are vague and would require either reference or additional explanation. These broad summaries may be removed. Below are examples.

L1-2: "Southern West Africa has one of the fastest growing populations worldwide. This has led to a higher water demand and lower air quality." No evidence and no clear reference. Fast population growth→ more energy demand→ more pollution can be discussed in the body of the paper. This reviewer did not find investigation on population increase and implication of this in the body of the paper.

*This statement sets the stage for our investigations and is an important motivation. We struggle to see why it would need additional explanation.*
*In the abstract we cannot include references, so statements like this will always stand alone. In the main body of the paper we cite the UN for population statistics, the World Bank report for water demand and Liousse et al. (2014) for air quality.*

E.g. Fig. 2 in https://www.tandfonline.com/doi/pdf/10.1080/02626660109492888

L3: "little" dry season. What does that mean? Is there "large" dry season? This is mainly a terminology issue. I encourage the authors to consider changing to an appropriate phrase. I am not sure if "short" could substitute "little". Terms "Short rain season, longer rain season, etc." are widely used in African monsoon literature.

*"Little dry season" is a widely used term in literature. The first prominent mentioning we found was in Griffiths (1972, Nigeria. World Survey of Climatology, H.E. Landsberg, Ed., Elsevier, 167–192) who cites Ireland (1962). Alternative terms like "secondary dry season" were used by Trewartha (1981), for example, but are used much less frequently. Thus, we prefer to stay with this phrase. The main dry season during boreal winter is mentioned at other places, see, e.g., your comment to Line 27.*

L5-7: "Increased pollution"→ "dimming solar radiation"→ "suppressing rainfall" – For a "linear" reader this would be hard to understand. I would encourage more clarity.

*We rephrased this part to make this more clear: "The proposed mechanism is that the dimming of incoming solar radiation by aerosol extinction contributes to reducing vertical instability and thus convective precipitation."*

L10-12: " ... decreases in horizontal visibility and incoming surface solar radiation are consistent with the hypothesized pollution impact . . ." . I would defer this to later sections. Or, authors may consider adding clarity.

*We prefer keeping the radiation and visibility aspects in the abstract as these are important parameters in our study. We added "…strong indicators for an increasing aerosol burden…" to add clarity.*

Sections 1-3:

Line 19-25: I encourage authors to consider to significantly reduce or improve the introduction. Most of the statements are not relevant. Just a couple of sentences to indicate the relationship between increasing population and energy demand and what that means to vulnerability. The overarching objective of the paper is explain rainfall trend and variability and the role of pollution resulting from this.

*We carefully restructured large parts of the Introduction to give it a clearer story and better flow. What is known about the overarching objective - rainfall trends in the region - is described much earlier in the second paragraph in the revised version. Some parts of the old second paragraph were removed or moved to later places in order to focus on the relevant statements. Answers to specific remarks are given below.*

L27: "Generally, the climate of SWA is strongly controlled by the seasonal evolution of the West African monsoon. Given the proximity to the equator, . . ..". This is vague. SWA is part of W. African monsoon (note: "Generally" is used). There are several examples that can show bi-modal, tri-modal rainfall away from the equator (about 10o away from Equator). As the authors know very well, the reason low-latitude regions exhibit bimodal or multimodal rainfall pattern are associated with north-south swing of the convective zone (ITCZ in the earlier literature). For example, L28 – dry season Dec-Feb. is related to the convective zone far to the south. I think the month-to-month variability should be described in that context and as described in along lines 31-34.

*In the revised introduction we deleted the term "Generally" and mention that the convective zone is further south during the dry season in boreal winter to avoid misunderstandings.*

L44: I recommend extra care in the use of adjectives or modifiers. What does "hardly organized convection" mean? Weakly organized? Also see comments above. Considering proximity of SWA to sea, and given the description in lines 35-45, what is the role of land-sea breeze?

*We replaced "hardly" by "weakly".*

*We also now mention the land-sea breeze in the climatological description in section 3:*

*"In contrast, during the SRS, the vertical profiles of shear and instability lead to a more local convective triggering, for example by the land-see breeze, and therefore smaller and shorter-lived but often intense systems (+ in Fig.\ 3)."*

*"Land-sea breeze days have a seasonal minimum (Guedje et al. 2019) and stability is often too high to allow the transition from cumulus clouds in the coastal hinterland into deep cold convective clouds in the course of the afternoon."*

Overall: the first few paragraphs of the Introduction may refocus on trend and mechanisms that potentially explain the trend and setting the stage for "pollution effect".

*The proposed mechanism is described much earlier in the revised version to give this topic more weight.*

L47: Again, I suggest remove "traditionally" and rephrase the sentence. Something like . . . " In the literature, marked SWA rainfall variability on interannual to decadal timescale is linked to . . .."

*We removed "traditionally" and kept the rest as the references at the end of the sentence make a phrase like "in the literature" obsolete.*

L50" ". . .undergone a mild recovery of rainfall since then . . ..". Can you be more specific? In the 1990? 2000s?

*Strong interannual variability does not allow to analyse the trends on very short time scales. Starting this sentence with "The 1980s stand out as a particularly dry period" clearly demonstrates that the recovery starts after this decade. Specifying this any further makes little sense in our opinion.*

L54-56: "While SST changes appear to have played a role in creating this trend, the seasonality and magnitude of changes remains poorly understood ..." Any reference? This inference is drawn after discussion of "interannual and decadal" variability and trend. It is unclear how is "seasonality" can be discussed in the same line(s).

*We rephrased this sentences and added a reference to make it clear that the long-term trends of the single seasons (FRS, LDS, SRS) remain poorly understood: "While SST changes appear to have played a role in creating this trend (Diatta & Fink, 2014), the magnitude of seasonal changes, i.e., the trends during the FRS, LDS and SRS, remains poorly understood."*

In the data and methods section, various techniques of trend analysis are discussed. Various parameters that have influence on rainfall have been explored. One way of investigating a quick and simple method to identify most important impact on a predictand is to run a partial correlation analysis. I'm not suggesting to redo the results, but I would like to know if this was considered. The reason I suggest this is that often times two predictors can have strong relationship and can affect one another and analysis based on linear relationship can give biased results.

*Indeed, we considered a partial correlation analysis but in the end decided not to do it for the following reasons:*

*As discussed in the manuscript and known from literature, the largest impact on rainfall variability comes from SSTs. Hence, we build a linear model that removes the SST-related variability in rainfall from the rainfall time series.*

*The other discussed parameters, aerosol, clouds, radiation, build a complex system in which all variables influence each other and are more or less correlated with each other. The results of a partial correlation analysis would point to the parameter that has the strongest correlation with rainfall. However, this parameter is again strongly influenced by the other players. Therefore, we first describe the proposed mechanism about the interplay of all the parameters. Then we focus on aerosol which we hypothesise to be to the most independent variable of the discussed ones, which stands at the beginning of our mechanism ("increased pollution" → "dimming solar radiation" → "suppressing rainfall") and which has other "external" drivers, such as local emissions and advection from remote regions.*

*We do not build predictors for rainfall from a combination of all the parameters. Their internal relationship, indeed, would give biased results.*

Association between AOD and rainfall: I would expect a timeseries plot that shows how radiation, cloud cover and horizontal visibility including relative humidity (RH) > 95% vary over time. Before discussing trends, a reader would have a chance to form an idea about how these parameters change over time. I believe this could be justified. This could be shown in section 3.1.

*Very good suggestion. We added new Figures 4, 8, and 11 in the revised version, which show the suggested time series and they are now discussed in detail when describing the trends of the respective parameter.*

Lines 275: Discussion can be made clearer if you refer to Fig. 3.

*We added the reference to Fig. 3.*

Yes, The RH >95% is in LDS deeper compared with other seasons. Deeper RH may be associated with stronger meridional winds (winds are also stronger at lower levels of the troposphere). So, more moist air is brought into (or circulating) the region from ocean. But, this may not be translating into rainfall. Would a local dynamic factor lacking? As shown, LDS is characterized by weak convective organization. Climatologically the convective zone is far to the north. Please explain.

*Here we discuss the RH>95% in the context of the wet growth of aerosol particles, not in the direct context of moisture source for rainfall. We rephrased the sentence to make this clear:*

*"…during the LDS the potential for wet aerosol growth is higher and that considerable aerosol effects could in fact be spread over a much larger region..."*

Line 280: Is 06UTC closer to cloud maximum? Since the solar time for the study area is almost the same as the Greenwich time, 06UTC is 6AM local solar time. If my assumption is true, cloudiness over SWA is more like the oceanic condition. Does information for cloud cover is more from more stratified clouds? I thought convective cloud coverage is later in the day for coastal areas (solar noon).

*06 UTC is closest to the maximum of low-level cloud cover which is crucial for the triggering of convection in the afternoon. Here we added some text and references in the revised version: "…analysis time closest to the diurnal maximum of low clouds (van der Linden et al., 2015). These clouds are crucial for the radiative energy budget and, thus, for vertical mixing and the triggering of convection in the afternoon (Kalthoff et al., 2018; Kniffka et al., 2019)."*

Lines 335-340: Question mark after a reference. Indicates something is missing.

*Corrected.*

In the Introduction section, there is a discussion related to aerosol transport from out of the study area. Although qualitative descriptions are provided, Aerosol transported from out of the region is not quantified or shown. In lines along line 340-350, since the trend can not be fully explained by the parameters, the conclusion was "anthropogenic aerosols play a role". This is unclear. How do we know whether anthropogenic aerosols play the main role in the rainfall trend? There could be multitude of factors that could explain the patterns. I guess the problem is the relationship between variables identified and aerosols not firmly established ("indirectly" to borrow authors words). I recommend adding a sentence or two explanation to help readers. Explain how "inland transport" might occur (in absence of data, literature might help).

*What we intended to state here was that these patterns are in line with our hypothesis (aerosol affects rainfall). We clarified this in the revised version: "This pattern reflects the regions where anthropogenic aerosol is assumed to have a major effect (close to the coast) and a minor or no effect, respectively (far inland and over hilly terrain) for the following reasons:…"*

Lines 378-382: This paragraph is interesting. The authors stated that MODIS AOD data is either "not useful" or very limited. Given this justification, I'm wondering why authors decide to continue using it? I suggest the rational for continuing the investigation is unclear. I encourage authors to re-write this.

*Well spotted! We focus on MODIS AOD over the Gulf of Guinea from where aerosol particles are assumed to be mainly transported into the study region with the predominantly southerly flow. We added this statement to the text.*

Section 3.3.2: Visibility and cloud cover As described in the text, the source of visibility measurements are observers. That means this data is prone to biases. One person's visibility estimation could be different form another person's estimate (different people taking observations over time). I am curious if authors consider a different way of looking at the same data. That is, (1) categorize visibility estimates into ranges, e.g., low, medium and high, then (2) workout the frequencies of each category. For example, what is the frequency of low visibility? Then, this can be used to describe the objective of the research. Higher frequency of low visibility can be related to high frequency of fog or bad weather, or pollution. Trend analysis can be performed on frequencies. The advantage of this approach is reducing biases because high and low visibility are categories and do not directly point to a measurement. Interpretation would also be easier. I thought about this because of the following!

I have hard time understanding a 200m/yr reduction in visibility (Line 420). What is the difference between an average 1km low visibility and average 0.8km visibility? How do this be a proxy for aerosols?

*Thank you for this suggestion. In the revised version we replaced the plots of trends of absolute visibility values with those of categorized visibility. We distinguish between visibility ranges of "up to 10 km", "10–20 km", and "above 20 km". The time series of the mean over all stations is shown in the new Fig. 8. Trends for all categories are shown for all stations separately in the revised Fig. 9. These new plots impressively illustrate the decrease in horizontal visibility in the region.*

Figures:

Captions are well explained but can be shortened at places. For example, in Fig. 3 2.0 and ERA5 can be used while omitting their full description. Meridional wind depiction is confusing. By plotting it as a vector, you are assuming zero zonal wind. Since you are considering one component of the wind field, you may consider using a line plot (with solid for positive and dashed for negative). Describe or give context to symbols just below season captions (e.g., ++ higher convection, O less, and + moderate)

*We shortened captions where suitable. With regard to the meridional wind component, we prefer to keep the illustration as a vector because plotting it as a line would require a second scale to show the strength. Together with the scale for RH>95% and cloud cover this would overload the picture. We added a sentence to the caption that a vector to the right (left) shows southerly (northerly) flow and added the description of the symbols.*

---

## Author Response (AR2)

Second review of the manuscript

"Increasing manmade air pollution likely to reduce rainfall in southern West Africa"

submitted to Atmos. Chem. Phys. Discuss., https://doi.org/10.5194/acp-2020-463-RC1

by Gregor Pante et al.

**Editor Decision: Publish subject to technical corrections** (10 Nov 2020)

by Ademe Mekonnen

I commend the authors for a comprehensive paper and a research area that is overlooked. However, I also agree with minor comments that are provided by Reviewer 1 regarding connection between air pollution and rainfall reduction. As Reviewer 1, a slight modification of the title would address concerns of the Reviewer and potentially readers. Authors suggested the word "likely" would address Reviewer's concern. Clearly, Reviewer 1 disagrees and so do I. Semantics aside (myself being non-native English speaker), I suggest something along " The potential impacts of increasing man-made air pollution on rainfall over ....". I encourage authors to think about Reviewer 1's concern.

*We understand the editor's and the reviewer's concerns about the title of the manuscript. The new title reads: "The potential of increasing man-made air pollution to reduce rainfall over southern West Africa".*

*We thank both the editor and the reviewer again for their comments!*